# Demographic and genetic factors influence the abundance of infiltrating immune cells in human tissues

Andrew R. Marderstein 1,2,3,4, Manik Uppal 2,3, Akanksha Verma1,2,3, Bhavneet Bhinder2,3, Zakieh Tayyebi 1,2,3, Jason Mezey1,2,4, Andrew G. Clark 1,4,5✉ & Olivier Elemento1,2,3,5✉

Despite infiltrating immune cells having an essential function in human disease and patients' responses to treatments, mechanisms influencing variability in infiltration patterns remain unclear. Here, using bulk RNA-seq data from 46 tissues in the Genotype-Tissue Expression project, we apply cell-type deconvolution algorithms to evaluate the immune landscape across the healthy human body. We discover that 49 of 189 infiltration-related phenotypes are associated with either age or sex ($FDR < 0.1$). Genetic analyses further show that 31 infiltration-related phenotypes have genome-wide significant associations (iQTLs) ($P < 5.0 \times 10^{-8}$), with a significant enrichment of same-tissue expression quantitative trait loci in suggested iQTLs ($P < 10^{-5}$). Furthermore, we find an association between helper T cell content in thyroid tissue and a *COMMD3/DNAJC1* regulatory variant ($P = 7.5 \times 10^{-10}$), which is associated with thyroiditis in other cohorts. Together, our results identify key factors influencing inter-individual variability of immune infiltration, to provide insights on potential therapeutic targets.

[1] Tri-Institutional Program in Computational Biology & Medicine, Weill Cornell Medicine, New York, NY, USA. [2] Institute of Computational Biomedicine, Weill Cornell Medicine, New York, NY, USA. [3] Caryl and Israel Englander Institute for Precision Medicine, Weill Cornell Medicine, New York, NY, USA. [4] Department of Computational Biology, Cornell University, Ithaca, NY, USA. [5] These authors contributed equally: Andrew G. Clark, Olivier Elemento. ✉email: ac347@cornell.edu; ole2001@med.cornell.edu

Human immune systems vary dramatically across individuals, yet the environmental and genetic determinants of this variability remain incompletely characterized. Previous studies have identified genetic components and environmental stimuli that alter immune cell composition in peripheral blood[1–6]. However, the immune system extends beyond peripheral blood, as infiltrating immune cells comprise a subset of the diverse cell types that compose bulk tissues and organs. The variability in this infiltration across individuals and between tissues has not been documented and the mechanisms enabling such variation in baseline infiltration have not been elucidated.

Identifying the genetic influences on specific patterns of infiltrating immune cells is crucial to understanding disease biology. Beyond further explaining heritable manifestations of infectious diseases and autoimmunity[1–9], such efforts can further uncover the drivers of characteristic immune cell signatures in the tumor microenvironment that are prognostic for cancer progression and predictive of treatment response[10]. For example, response is improved in patients with T-cell-inflamed tumors compared to T-cell-depleted tumors among patients receiving immune checkpoint inhibitors targeting PD-1 and CTLA-4[10,11] and among ovarian cancer patients receiving chemotherapy[12]. However, a complete mechanistic description underlying immune-rich and immune-poor tumor phenotypes remains elusive.

Recent advances in computational methods have allowed reliable inference of the heterogeneous cell types from gene expression data of a single-population-level (bulk) tissue sample[13–15]. At the same time, large-scale sequencing efforts such as the Genotype-Tissue Expression (GTEx) project[16] have enabled a detailed exploration of the links between genomic and transcriptomic variations across different tissues. Together, these cell-type estimation methods can be utilized in synergy with massive bulk sequenced data sets to infer cellular heterogeneity and achieve statistically well-powered associations that uncover the intrinsic drivers of heterogeneity[17–19].

In the present study, we aim to evaluate the inherent immune infiltration landscape across healthy tissues in the human body and to determine intrinsic factors contributing to the infiltration variability. We use bulk RNA-seq data from 46 distinct GTEx tissue types and cell-type deconvolution algorithms under a new analysis framework to leverage information across different deconvolution methods for association testing. We identify age-related and sex-related associations with tissue immune content and perform a genome-wide association study (GWAS) analysis to discover germline genetic effects. We identify several genetic and non-genetic associations with immune cell patterns, which can serve as leading candidates for understanding the basic biology behind tissue infiltration.

## Results

### Estimation of immune cell types in bulk RNA-seq profiles. To describe immune content from bulk RNA-seq samples, we used two central algorithms: xCell[13] and CIBERSORT[14]. Both algorithms include slightly different cell types and reference gene sets for estimation. xCell relies on a modification of single sample gene-set enrichment analysis to estimate cell-type scores of 64 immune and stroma cell types, including various subtypes of CD8+ T cells, CD4+ T cells, B cells, dendritic cells, macrophage polarization states, and other innate immune cells. CIBERSORT employs a linear support vector regression model to estimate cell-type "relative" proportions of 22 immune cell types. This includes many of the same broad cell types as xCell, but with fewer subtypes (see Methods section for the list of cell types estimated by xCell and CIBERSORT that were used in this manuscript). Additionally, CIBERSORT calculates a "scaling factor" to measure the amount of total immune content in the sample, allowing the

calculation of "absolute" scores (which are the product of the scaling factor and cellular proportions). We refer to the relative proportions from CIBERSORT as "CIBERSORT-Relative" and the product of the relative proportions with the scaling factor as "CIBERSORT-Absolute". Using xCell, CIBERSORT-Relative, and CIBERSORT-Absolute, we estimate three scores for each cell type to describe the immune content underlying gene expression data.

We first hypothesized that the relative and absolute scores from CIBERSORT encapsulated different aspects of the cell-type deconvolution, which can provide alternative perspectives of the tissue-immune environment. While "CIBERSORT-Absolute" simultaneously quantifies the amount of immune content, "CIBERSORT-Relative" is purely focused on capturing compositional changes in the immune content (Supplementary Note 1). We simulated synthetic mixes composed of bulk tissue "spiked" in silico with CD4+ T cells and CD8+ T cells (see Methods). We correlated the known amount of CD4+ and CD8+ T cell infiltration in these mixtures with estimated deconvolution scores under a "tissue" scenario and an "immune cell" scenario. In the "immune cell" scenario, we let the true infiltration be the proportion of each cell type to the total immune content. In the "tissue" scenario, the true infiltration amount is the proportion of each cell type to the entire sample. As expected, we found CIBERSORT-Relative to more accurately estimate the infiltration amounts of the in silico mixtures than CIBERSORT-Absolute in the "immune cell" scenario, while the reverse was true in the "tissue" scenario (Supplementary Table 1). However, in both scenarios, the CIBERSORT method resulted in strong correlations between deconvolution scores and the true amount of infiltration ($r = 0.64-0.90$).

Additionally, we were interested in comparing the performance of CIBERSORT to xCell. Since xCell is an enrichment-based algorithm, not a deconvolution algorithm, it is not recommended for comparing scores between cell types. As a result, xCell scores correlated well in the "tissue" scenario ($r = 0.90$ with CD4+ T cells, $r = 0.96$ with CD8+ T cells) but worse in the "immune cell" scenario ($r = 0.65$ with CD4+ T cells, $r = 0.45$ with CD8+ T cells), even after normalization (Supplementary Table 2; compared to CIBERSORT). We also found that xCell had imperfect correlation with CIBERSORT-Absolute scores (CD8: $r = 0.58$, CD4: $r = 0.86$). Therefore, xCell provided an accurate estimation of a tissue's immune content that is distinct from CIBERSORT.

Thus, our analyses revealed that relative scores better captured compositional differences in immune content, while absolute scores better captured the true cell-type amount to the overall sample. Furthermore, while xCell-based and CIBERSORT-based estimates correlate with true immune cell amounts and with each other, there is not perfect correlation (see Supplementary Figs. 1 and 3 for case examples where xCell and CIBERSORT differ in simulated data and empirical data respectively). As a result, an effect or difference may be better captured and detected in one deconvolution method compared to another. These observations indicate that it could be favorable to leverage information across deconvolution estimates. Furthermore, they suggest that the optimal analysis strategy will jointly consider CIBERSORT-Relative, CIBERSORT-Absolute, and xCell scores, equally but independently, to best capture the full range of deconvoluted immune content (see Supplementary Note 2).

### Evaluating infiltration across human tissues. Our next objective was to investigate the immune landscape across the human body by deconvoluting cellular heterogeneity within bulk RNA-seq GTEx samples (Fig. 1a). The v7 GTEx release consists of 11,688 samples, spanning 53 different tissues/sample types from 714 donors[16]. We performed comprehensive deconvolution of $n = 11,141$ samples from 46 tissue types using our three methods

**a** 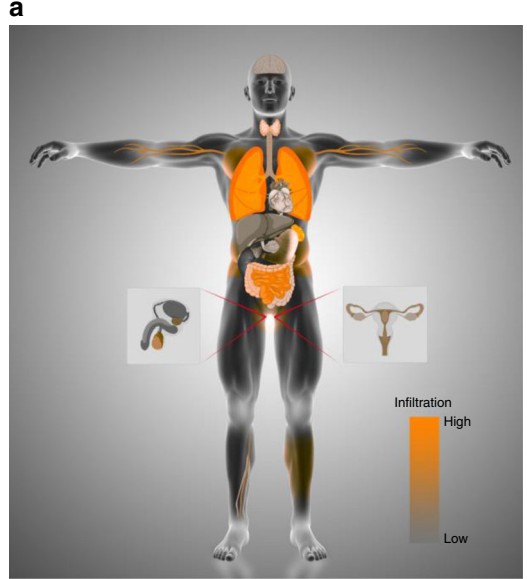

**b** 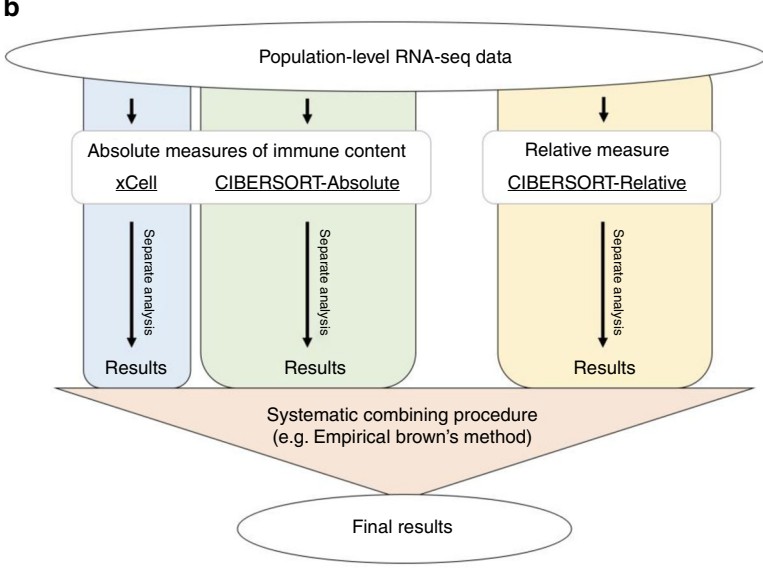

**Fig. 1 Study overview. a** Using deconvolution algorithms and population-level transcriptomes, we estimated the amount of specific immune cell types in 11,141 samples from the Genotype-Tissue Expression project. This data set spans 46 distinct tissue types. We visualize the total amount of infiltration, as calculated by the scaling factor in CIBERSORT-Absolute, in each of these tissues. Source data are provided as a Source Data file. **b** A graphical overview of downstream statistical analysis. Immune content was estimated by using three different algorithms. xCell estimates various subtypes of CD8+ T cells, CD4+ T cells, B cells, dendritic cells, macrophage polarization states, innate immune cells, and non-immune cells. CIBERSORT, which only measures immune cells, estimates fewer subtypes and instead distinguishes between resting and activation states of major cell types. Both algorithms utilize different reference gene sets. Statistical analyses were performed on each of these three estimates separately. The final results were obtained by combining the different separate analyses.

(xCell, CIBERSORT-Relative, and CIBERSORT-Absolute), estimating the amount of 14 distinct immune cell types in each sample (CD8+ T cells, CD4+ naive T cells, CD4+ memory T cells, helper T cells, regulatory T cells, gamma delta T cells, B cells, NK cells, neutrophils, macrophages, dendritic cells, mast cells, monocytes, and eosinophils; Fig. 1b; see Methods and Supplementary Tables 3 and 4). We note that deconvolution of the true GTEx gene expression profiles produced very distinct estimates compared to "control" samples, where the gene expression profile is randomly shuffled (see Supplementary Note 3 and Supplementary Fig. 2).

To obtain an overview of immune content across the human body, we first performed hierarchical clustering of the tissues based on estimated immune content. In brief, we clustered the tissues based on the median values of each cell type across individuals (see Methods). Many nearest-neighbor pairings were consistent across all three deconvolution outputs and recapitulated relationships between tissues that share high degrees of histologic similarity and immune infiltration (Fig. 2a, Supplementary Fig. 4). For example, sun-exposed and non-sun-exposed skin samples, gastroesophageal junction and muscularis esophagus samples, and several brain sub-tissues cluster closely. Interestingly, we observed that transverse colon, sigmoid colon, and small intestine (terminal ileum) tissues cluster distinctly, with a decrease in T cell content from the small intestine to the rectum that corroborates previous clinical observations[20]. Further analysis, such as in macrophages, emphasized the vast differences in immune content between distinct tissues (Fig. 2b). We found the highest macrophage scores in tissues are where resident macrophages are well-characterized, such as in adipose and lung samples. In contrast, much lower macrophage content was estimated in testis and many brain sub-tissues (Supplementary Notes 4 and 5).

Importantly, we found large variability among individuals within a single tissue (Fig. 2b). Many tissues featured a majority of samples with a minimal amount of immune cells, but these tissues also contained several samples with significantly higher estimated immune content (Supplementary Figs. 6–9). Interestingly, t-SNE visualizations of estimated immune content within a tissue type do not reveal clear clusters of samples (Supplementary Fig. 10). Therefore, it appeared that healthy individuals have highly variable infiltration patterns, suggesting that the differences could be driven by a range of genetic and non-genetic factors (such as age, sex, or environmental exposures). We focused on searching for these factors in a limited set of 189 filtered infiltration phenotypes, which represent the amount of a particular immune cell type in a specific tissue on a continuous scale. This set was derived from an unfiltered list of 736 infiltration phenotypes, which encompass 14 specific immune cell types (described above) and 2 broader cell types (lymphoid and myeloid) in the 46 GTEx tissues, as estimated across three separate measurements (by CIBERSORT-Relative, CIBERSORT-Absolute, and xCell). The set was filtered down to 189 phenotypes, which was performed using several criteria that considered sufficient immune content and correlated estimations between xCell and CIBERSORT (Methods section and Supplementary Data 1).

Using the filtered set of 189 infiltration phenotypes, we were interested whether infiltration signatures explain substantial variance of gene expression calculated in bulk assays. We performed a principal component analysis of the processed gene expression matrix within each tissue, before assessing the pairwise relationship between the first four principal components and the 189 infiltration phenotypes using CIBERSORT-Absolute scores. Even after a Benjamini-Hochberg false discovery rate (FDR) correction[21], we found that 183/189 infiltration phenotypes were significantly correlated with at least one principal component (FDR < 0.1). This indicated that a significant proportion of the variance in bulk gene expression measurements are due to the cellular heterogeneity driven by immune content (Supplementary Data 2).

**Identification of extreme infiltrating immune cell patterns.** As the variance in the bulk gene expression profiles was partly

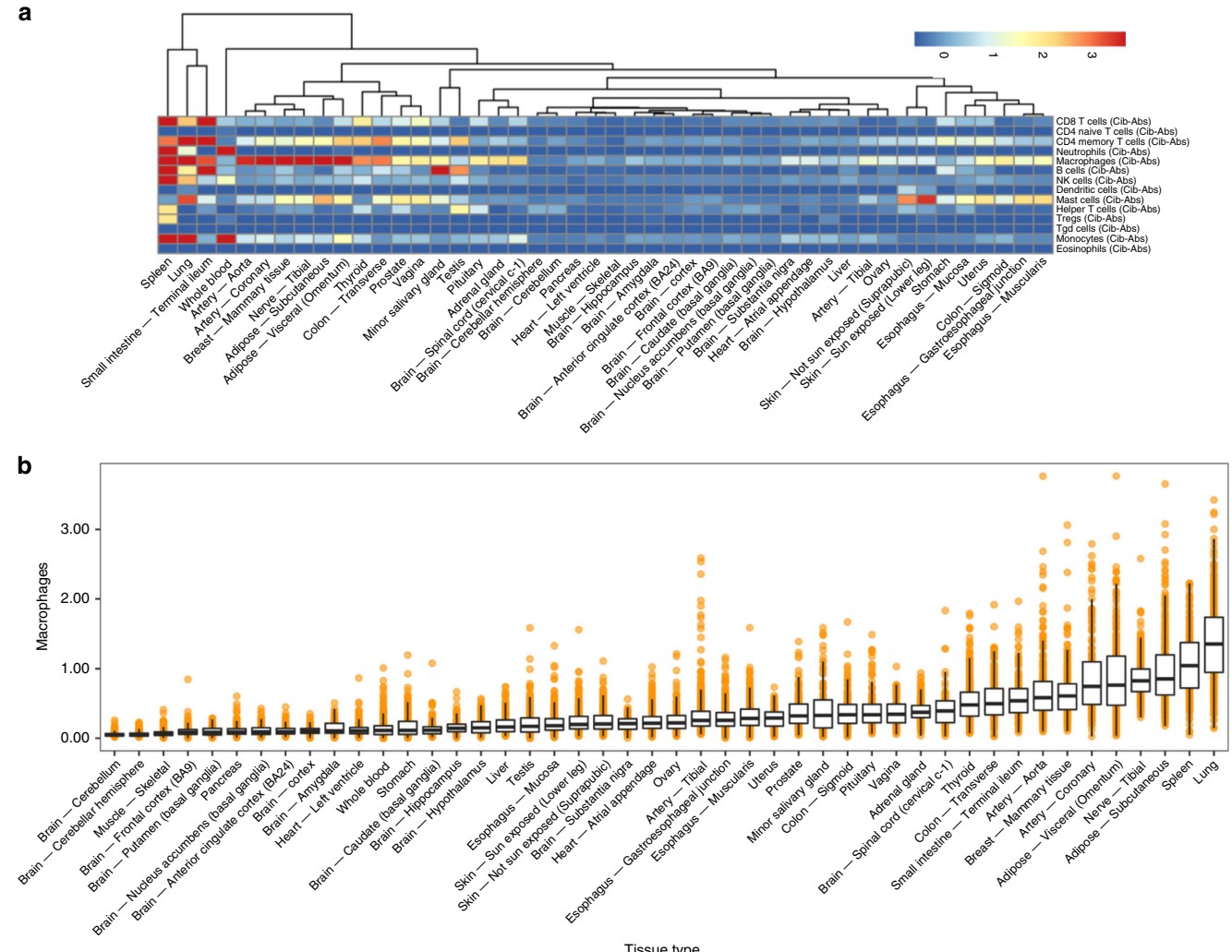

**Fig. 2 Evaluating immune content across tissues.** The quantity of 14 different immune cell types were estimated in GTEx samples using xCell, CIBERSORT-Absolute, and CIBERSORT-Relative deconvolution. **a** For CIBERSORT-Absolute, the median scores for each cell type were calculated within each tissue. Hierarchical clustering was performed on tissues. The dendrogram and standardized median cell type scores were visualized in a heatmap. **b** CIBERSORT-Absolute macrophage scores were plot for $n = 11{,}141$ samples across 46 tissue types. The tissues were sorted by median score. Data is summarized as boxplots where the middle line is the median, the lower and upper hinges represent the first and third quartiles, and the whiskers extend from the hinge with a length of 1.5x the inter-quartile range. All data points are plot individually. Source data are provided as a Source Data file.

explained by immune cell content, we next aimed to characterize the primary transcriptomic differences among samples with vastly different immune content. We explored two approaches for defining immune cell type-rich (hot) versus immune cell type-depleted (cold) cases for a particular infiltration phenotype. In the "quintile" approach, we characterized all samples with scores consistently in the highest 20% ("hot") and lowest 20% ("cold") across all three deconvolution methods. In the "consensus clustering" approach, we used consensus k-mean clustering to identify the immune-rich (hot) and immune-depleted (cold) samples (Methods section; graphical methods in Fig. 3a; Supplementary Figs. 11 and 12; Supplementary Data 3).

Using these clusters, we performed differential expression analysis. We found that the two cluster characterization approaches yielded similar differentially expressed genes, such as many well-known markers of immune cell types present in the reference gene sets of xCell and CIBERSORT. After filtering genes included within either reference gene set, the most common differentially expressed genes largely corresponded to other immune cell type genes (that were not used by the xCell and CIBERSORT algorithms), such as various cytokines or

immunoglobulins. To identify key pathways from our differentially expressed genes, we used Ingenuity Pathway Analysis. This identified immune signaling, immune cell maturation, and inflammation pathways as the most commonly dysregulated across infiltration phenotypes (see Supplementary Notes 6–10).

Finally, we used our immune-hot clusters (e.g. macrophage-hot) to examine whether individuals with inflammation in one tissue type may also exhibit similar inflammation in their other tissue types. For each cell type, we analyzed the distribution of hot tissues across individuals with at least eight different tissue samples. Within the consensus clusters, we discovered that individuals were labeled hot in an average of 9.1–12.5% tissue samples per cell type, with a mode of one hot tissue per individual for a single cell type (Supplementary Fig. 13; within quintile clusters, individuals were labeled hot in an average of 16.6–25.0% tissue samples). In addition, across individuals, there were no clear, common hot inflammation patterns representing multiple tissues (Fig. 3b; Supplementary Fig. 14). We developed a statistical method to formally test the hypothesis of independent hot inflammation patterns between any two tissues (Methods section). Out of 1796 tissue pairs tested, this hypothesis could not be rejected for any

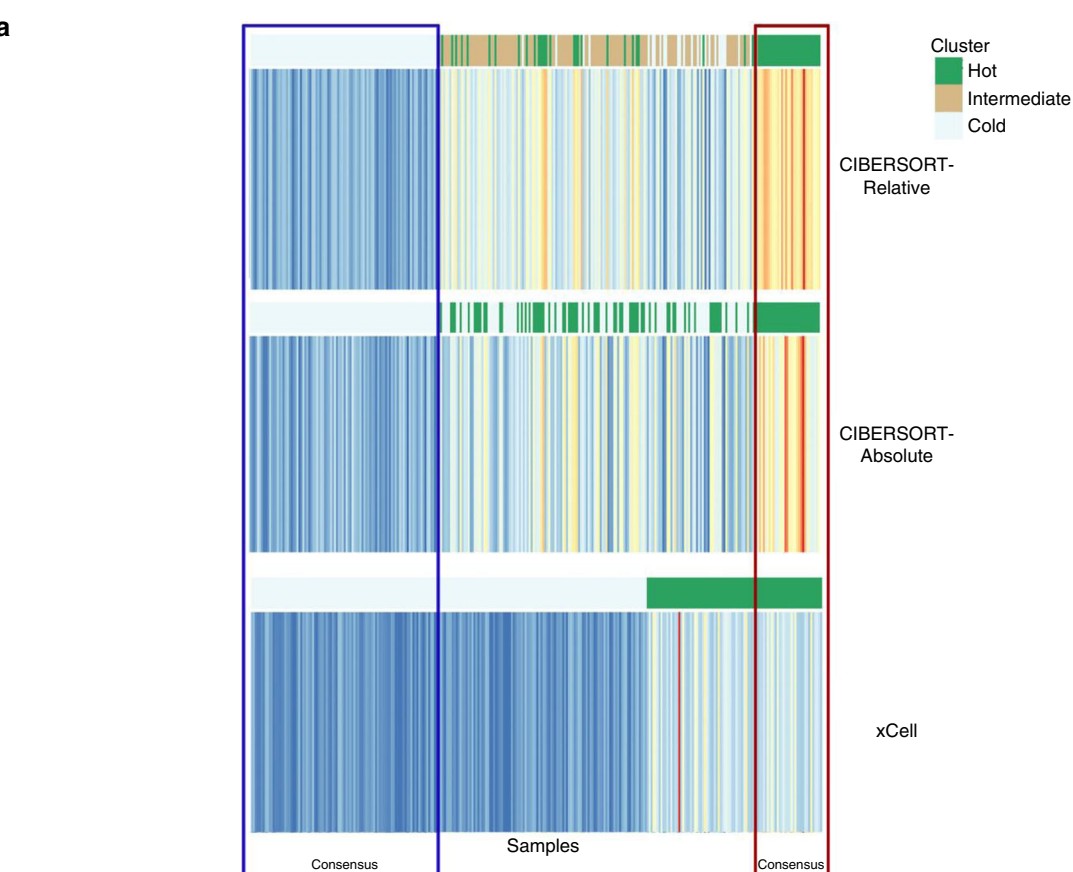

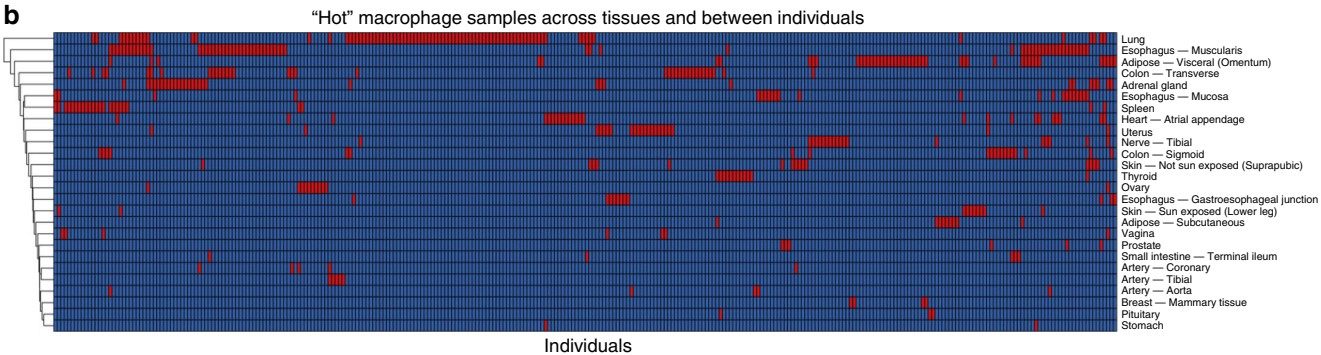

**Fig. 3 Identification of hot and cold infiltration patterns. a** Stacked heat maps depicting an infiltration phenotype: the immune cell scores for a particular tissue, as estimated across the 3 deconvolution algorithms. Samples are in columns, with the ordering of samples along the x-axis identical for the 3 heatmaps. Blue, yellow, and red colors correspond to low to moderate to high estimated scores. Annotations assigning samples to hot, cold, and intermediate clusters are depicted along the top of each heatmap. In this example, cluster assignments were determined using consensus k-means clustering. Samples identified as hot or cold across each of the 3 deconvolution methods were identified as the "consensus" hot and cold groups, and considered for differential gene expression analysis. **b** Heatmaps display hot patterns across tissues and individuals. Clusters were determined by the consensus k-means clustering approach. Rows represent tissues and columns represent individuals. Red indicates the individual was labeled hot in that tissue type, while blue represents not hot (intermediate, cold, or missing data since an individual does not have a sample for every tissue type). Row and columns were clustered by Euclidean distance. Source data are provided as a Source Data file.

tissue pair using the consensus k-means clusters ($P < 2.8 \times 10^{-5}$); however, by using the quintile-based clusters, we found evidence ($P < 2.8 \times 10^{-5}$) for an association between hot lung and hot whole blood samples for CD8+ T cell content ($P = 2.7 \times 10^{-6}$; in the consensus clustering analysis, we found a $P$-value $= 1.9 \times 10^{-3}$; Supplementary Fig. 15). We next relaxed the hot cluster requirements to include individuals within the top two quintiles (40%) across all three deconvolution methods, which comprise individuals at above average but not necessarily extreme immune

cell levels. Using these new clusters, we found nine pairs of tissues with significant associations ($P < 2.8 \times 10^{-5}$) for particular immune cells (out of 1796 tissue pairs tested; see Supplementary Table 5). Therefore, we note that extreme infiltration patterns appear to generally be phenotypically tissue-specific, rather than widespread (e.g. hot-sharing between tissues). However, when assumptions were relaxed to reflect above average rather than extreme immune content, there appeared to be evidence for particular pairs of tissues with some level of shared immunity.

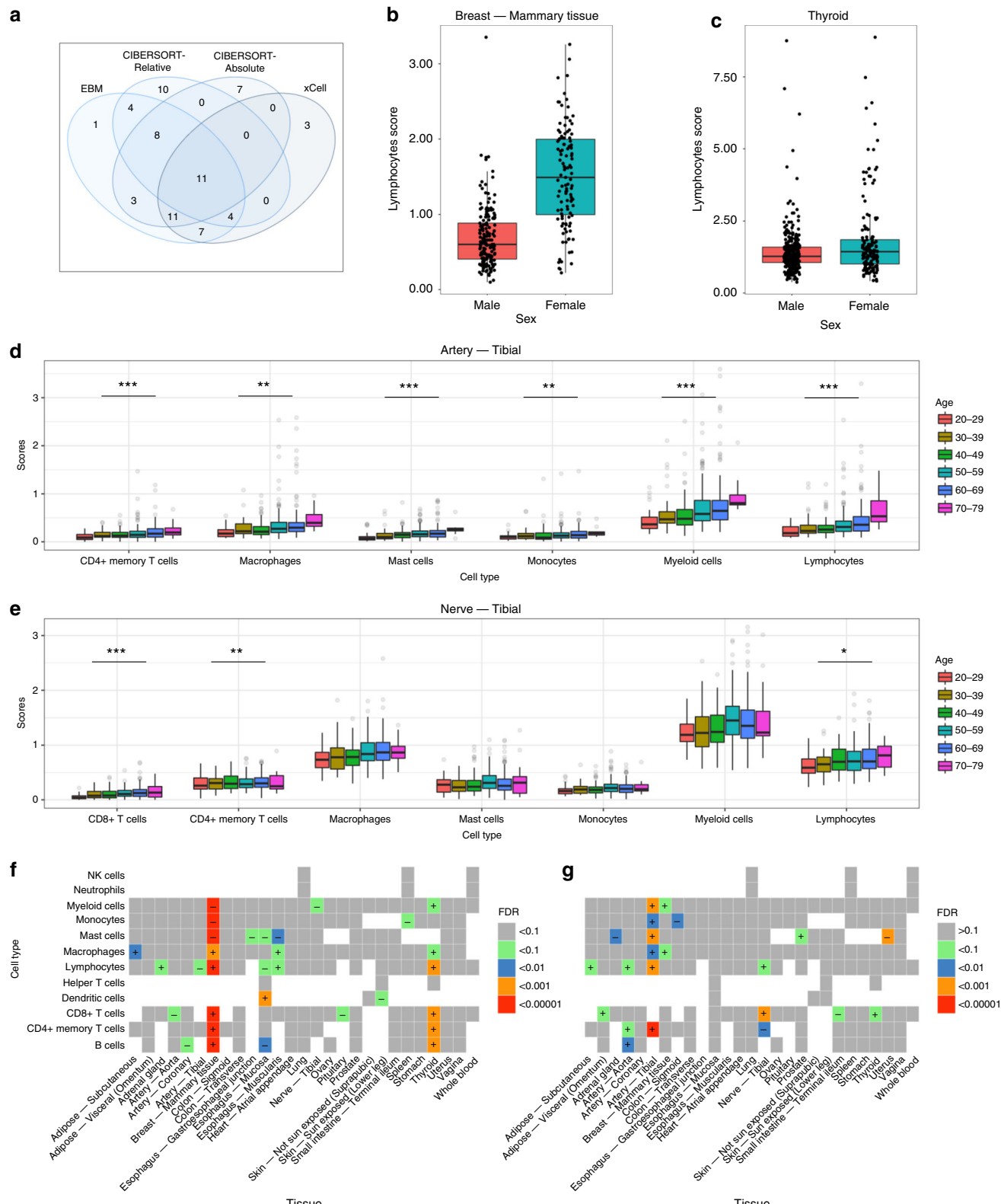

**Association of age and sex with immune infiltration.** We next aimed to examine whether there are any donor characteristics which influence the amount of immune cells observed in bulk tissue. We adopted a multiple regression approach to measure age and sex effects across all infiltration phenotypes. This was repeated for each of the deconvolution procedures, and we aimed to capture shared signals across the methods by merging P-values from all three with

Empirical Brown's method[22] (Fig. 1b; Methods section). In simulations from our synthetic samples, we found that combining P-values using Empirical Brown's method improved statistical power while maintaining the false positive rate (see Supplementary Note 2).

Overall, we observed that 49 out of 189 infiltration phenotypes (25.9%) were significantly associated with either age or sex (FDR < 0.1; Fig. 4f, g; Supplementary Data 9), with at least one

**Fig. 4 Significant associations with sex and age. a** Venn diagram displaying number of phenotypes associated with either age or sex by each analysis method (*FDR* < 0.1). EBM represents combined results using Empirical Brown's method. **b**, **c** The relationship between lymphocytes and sex in **b** n = 290 breast and **c** n = 446 thyroid samples. **d**, **e** The relationship between various immune cell types and age in **d** n = 441 tibial nerve and **e** n = 414 tibial artery samples. Significance is indicated by number of asterisks. One asterisk indicates *FDR* < 0.1, two asterisks indicates *FDR* < 0.01, three asterisks indicates *FDR* < 0.001. Displayed immune cell type scores in **a–d** are from CIBERSORT-Absolute. **f**, **g** Summary of (**f**) sex and (**g**) age association results from all 189 infiltration phenotypes. ± indicates effect direction from CIBERSORT-Absolute analysis (increase in females or higher age), and each cell is colored by FDR significance. Blank boxes are removed phenotypes (were filtered out). In **b–e**, data are summarized as boxplots. The middle line is the median, the lower and upper hinges represent the first and third quartiles, and the whiskers extend from the hinge with a length of 1.5x the inter-quartile range. In **b**, **c**, all data points are plot. In **d**, **e**, only data points beyond the whiskers are plot. All significance values are FDR-adjusted *P*-values of the coefficients in an age-sex multiple linear regression model. Source data are provided as a Source Data file.

immune cell type significantly associated with age or sex in 19 of the 27 analyzed tissue types. We found that the combined results for our age and sex testing resulted in an increased number of significant phenotypes compared to separate analysis (37 significant phenotypes using only CIBERSORT-Relative, 40 using CIBERSORT-Absolute, and 36 using xCell) by capturing shared signals across the methods (Fig. 4a).

We identified a greater number of associations with sex than age (phenotypes with *FDR* < 0.1: 22 for age, 31 for sex), with the most striking findings being increased lymphocytes in female breast and thyroid tissues compared to males (breast: $P = 6.6 \times 10^{-45}$; thyroid: $P = 1.3 \times 10^{-6}$; Fig. 4b, c). In female breast tissue samples, we observed significant heterogeneity, with several samples having few lymphocytes detected and others having high predicted lymphocyte content. The distinct contrast between female and male mammary breast tissue could drive these immune differences, including male tissue predominantly lacking the lobular elements[23] and the increased exposure to infections (e.g. mastitis) in females that are both associated with T cells (Supplementary Note 11 and Supplementary Fig. 16). Women also feature a higher prevalence of autoimmune and neoplastic thyroid disorders than males[24,25]. With age, the most significant associations were localized to the tibia area (nerve and artery tissues). Furthermore, 9 of 13 tibial-area infiltration phenotypes were significantly associated with age (*FDR* < 0.1), with the exception being myeloid-based phenotypes in tibial nerve samples (Fig. 4d, e). In comparison, only 3 of 16 phenotypes in artery tissue from other body areas (coronary and aorta) were significantly associated with age.

Lastly, while we identify many new sex- and age-associated changes in tissue immunity, an analysis of aged blood samples support previous findings from other studies, including myeloid-biased differentiation[26] (Supplementary Fig. 17), a rise in NK cells[27,28], and a decline in the ratio of CD4 to CD8 T cells[29] (defined as all CD4 and CD8 bearing T cells; Methods section and Supplementary Note 12 for results).

**Genetic variants associated with infiltrating immune cells**. We next searched for particular inherited genetic variants that could influence the variability of infiltration patterns. We refer to germline single-nucleotide polymorphisms (SNPs) associated with the amount of a cell type in a tissue, or an infiltration phenotype, as infiltration quantitative trait loci (iQTLs). Using the Empirical Brown's testing framework across 189 infiltration phenotypes, we discovered 31 infiltration phenotypes with at least one genome-wide significant iQTL ($P < 5.0 \times 10^{-8}$) in 15 out of 27 tissues (Fig. 1b; Methods section and Supplementary Data 10). The number of significant phenotypes in the combined analysis is double the amount as detected by any separate analysis (15 significant phenotypes using only CIBERSORT-Relative, 16 using CIBERSORT-Absolute, and 16 using xCell; Supplementary Data 10; Supplementary Note 13). While Fig. 5a and Supplementary Fig. 18 demonstrate the power of our Empirical Brown-

based approach to leverage statistical signals across the three deconvolutions, Fig. 5b shows that genome-wide significance is rarely achieved in multiple separate single-deconvolution analyses. Furthermore, we found that the top 31 iQTLs from these 31 phenotypes were significantly enriched for being a previous GWAS association: 19.4% of the 31 iQTLs have a phenotype association in the phenoscanner GWAS database of $P < 5 \times 10^{-8}$, compared to a 5.4% expectation (3.6-fold enrichment, permutation-based *P*-value = $5.5 \times 10^{-3}$; Methods section).

The most significant iQTL we identified was an association between rs6482199 and helper T cells (in particular, Th1, Th2, and T follicular helper cell content inferred by the deconvolution algorithms) in thyroid samples ($P = 7.5 \times 10^{-10}$; Fig. 5c, d). We conducted simulations to examine the false positive rate between this SNP and the phenotype, but found no evidence for *P*-value inflation (see Supplementary Notes 14 and 15). While this intergenic variant has not been linked to the expression of any gene in any tissue within the GTEx consortium analyses[16], it has been associated with *DNAJC1* and *COMMD3* gene expression in whole blood through the eQTLGen meta-analysis ($P = 3.6 \times 10^{-16}$ and $P = 5.8 \times 10^{-6}$ respectively)[30]. *DNAJC1* encodes a member of the heat shock family proteins (*hsp*), which are well characterized in stress and immune responses[31,32], and its transcripts and proteins are highly expressed in thyroid samples[33,34]. *COMMD3* proteins have been associated with immunity through interferon stimulation[35], regulation of NF-kappa-B activity[36], and lymphocyte migration by recruitment of specific G protein-coupled receptor kinases[37]. In the GTEx thyroid samples, we found that *DNAJC1* and *COMMD3* gene expression both correlated with the helper T cell phenotype ($r = 0.12 \pm 0.09$, $P = 0.01$ and $r = 0.31 \pm 0.08$, $P = 1.1 \times 10^{-11}$ respectively; Fig. 5e).

We were next interested in whether the genetic effects of this variant are associated with disease. We found that rs6482199 has been previously associated with both chronic lymphocytic thyroiditis in the Michigan Genomics Initiative participants ($P = 0.012$)[38] and self-reported thyroiditis in UK Biobank ($P = 9.6 \times 10^{-3}$)[39]. It has also been linked to numerous white blood cell counts[39] and glutamine measures[40] crucial for immune cell proliferation and function. Furthermore, this variant is in a potentially active regulatory region of the non-coding genome near enhancer histone marks and DNAse sites across various hematopoietic and non-hematopoietic cell types[41,42]. Thus, we next used GeneHancer[43] to query 49 other common SNPs (*MAF* > 0.01) located in 6 promoter and enhancer regions linked to both the *DNAJC1* and *COMMD3* genes. We then tested for association with self-reported thyroiditis in UK Biobank using the Neale lab analysis (N = 361141)[39]. Under Bonferroni significance ($P < 1.1 \times 10^{-3}$), we identified rs56186224, a variant located in an enhancer linked to both *COMMD3* and *DNAJC1*, to be associated with self-reported thyroiditis ($P = 4.2 \times 10^{-4}$). However, we could not test for infiltration effects at this SNP due to lower frequency in our GTEx data (there were only 11 alternate alleles

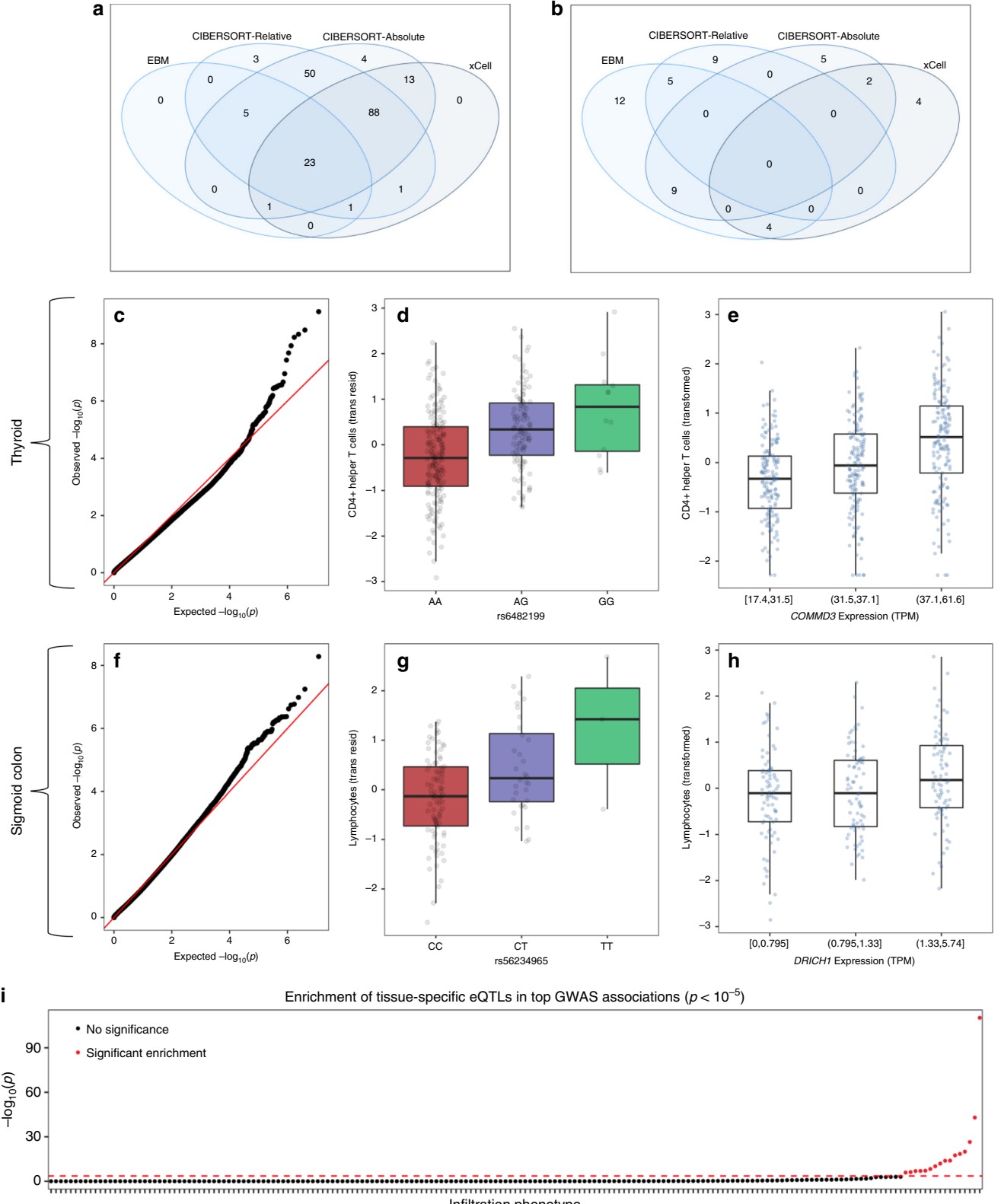

available). Nonetheless, our genetic results point to a potential role for *DNAJC1* and *COMMD3* in invasive thyroid inflammation.

The second-most significant iQTL we discovered was an association between rs56234965 and lymphocytes in sigmoid colon samples ($P = 5.2 \times 10^{-9}$; Fig. 5f, g). The variant lies within the intron of *DRICH1* (also known as *C22orf43*) and has been identified in GTEx as a multi-tissue *DRICH1* eQTL (all but two

GTEx tissues have a posterior probability that an effect is shared in each tissue > 0.9). A recent CRISPR knockout of *DRICH1* demonstrated its essentiality in human pluripotent stem cells, with severe proliferation defects and major transcriptional changes (including TGF-$\beta$ signaling and genes involved in cell fate decisions and differentiation)[44]. In UK Biobank, gene-wide *DRICH1* variants were associated with several intestinal death

**Fig. 5 Significant results in GWAS across 189 infiltration phenotypes.** The Venn diagrams in **a** and **b** display the number of infiltration phenotypes with a significant variant identified within each analysis procedure. **a** Empirical Brown's Method: $P < 5 \times 10^{-8}$. xCell, Cibersort-Relative, and Cibersort-Absolute: $P < 10^{-5}$. **b** All $P < 5 \times 10^{-8}$. Figures **c–h** relate to the two most significant genetic associations. In all, 5+ million genome-wide genetic variants were tested for association with Helper T cells in thyroid samples (top row) and lymphocytes in sigmoid colon samples (bottom row). The SNP-association Empirical Brown's p-values are visualized along the y-axis in a qq-plot (leftmost column), where the x-axis represents the expected −log10 p-values under the null distribution. The association between the most significant variant from the GWAS and the phenotype (using the transformed residuals) is visualized in the middle column. The right column displays the association between the phenotype (transformed raw scores) and tissue gene expression of variant-associated eGenes, with expression values split into three groups in the 33rd and 67th percentile of expression. For visualization purposes, CIBERSORT-Absolute scores are displayed in the middle and right columns. In **d**, **e** and **g**, **h**, data are summarized as boxplots where the middle line is the median, the lower and upper hinges represent the first and third quartiles, and the whiskers extend from the hinge with a length of 1.5x the inter-quartile range. All data points are plot individually. Due to non-present genetic data for some donors between v6 and v7 releases, $n = 281$ samples are used in **d** and $n = 446$ samples are used in **e**. Similarly, $n = 121$ samples are used in **g** and $n = 233$ samples are used in **h**. Figure **i** describes the statistical test results for whether there is an enrichment of eQTLs in the iQTL genetic results for each infiltration phenotype, where infiltration phenotypes are sorted along the x-axis by the enrichment test's −log10 p-values. Source data for **a**, **b**, **e–i** are provided as a Source Data file.

| Table 1 Summary of important terms defined in the study. | |
|---|---|
| **Term** | **Definition** |
| Infiltration phenotype | Estimated quantity of an immune cell type in a tissue |
| iQTL | A genetic variant associated with an infiltration phenotype |
| eQTL | A genetic variant associated with gene expression |
| ieQTL | A genetic variant that is both an eQTL & an iQTL in the same tissue |

causes (diverticular disease with perforation and abscess, $P = 2.9 \times 10^{-10}$; gastro-intestinal hemorrhage, $P = 1.6 \times 10^{-9}$; acute vascular disorders of the intestine, $P = 3.5 \times 10^{-6}$) and lympho-cytic cancer-related deaths (unspecified T-cell lymphoma, $P = 7.9 \times 10^{-19}$; unspecified non-hodgkins lymphoma, $P = 1.9 \times 10^{-6}$; chronic lymphocytic leukemia, $P = 4.4 \times 10^{-6}$). By analyzing gene expression in the sigmoid colon samples and the lymphocyte phenotype, we discovered a significant correlation ($r = 0.21 \pm 0.12$, $P = 8.9 \times 10^{-4}$; Fig. 5h). Furthermore, eQTL-Gen[30] analyses identified several other eQTL associations of rs56234965 in whole blood (in addition to *DRICH1* expression ($P = 10^{-167}$)): with *IGLL1/CD179B* expression ($P = 3.2 \times 10^{-7}$; a B cell surface receptor), *RGL4* expression ($P = 1.7 \times 10^{-20}$; encodes a protein associated with T-cell lymphomas[45] that may be able to activate the Ras-Raf-MEK-ERK cascade[46]), and *ZNF70* expression ($P = 4.8 \times 10^{-7}$; a regulator of HES1, a key transcription factor in regulatory programming and cellular differentiation[47]).

**Investigation of iQTLs in the context of eQTLs.** Variants already associated with gene expression, such as rs56234965 with *DRICH1*, allow inference of possible functional roles. Thus, we next investigated whether there were expression quantitative trait loci (eQTLs) from the GTEx consortium analysis[16] that were also iQTLs within the same tissue (ieQTLs) (see Table 1).

First, we identified one other infiltration phenotype with an ieQTL surpassing genome-wide significance ($P < 5.0 \times 10^{-8}$): rs9989443 with both *CCDC40* expression ($P = 4.2 \times 10^{-7}$) and mast cell infiltration ($P = 3.6 \times 10^{-8}$) in esophagus (muscularis) tissue. Previously, this variant had been associated with numerous immune-related proteins (such as leptin receptor and transforming growth factor-beta-induced protein ig-h3) in the INTERVAL study ($P = 1.7 \times 10^{-4}$ and $P = 4.7 \times 10^{-3}$ respectively)[48]. Gene-wide *CCDC40* variants have also been associated with myeloid leukemia death ($P = 1.7 \times 10^{-8}$) and self-reported esophagus disorders ($P = 1.8 \times 10^{-9}$) in UK Biobank[39]. Additionally, we discovered *CCDC40* RNA expression in esophagus (muscularis) samples to be correlated with the mast cell phenotype ($r = 0.25 \pm 0.1$, $P = 6.2 \times 10^{-7}$).

Next, we wanted to systematically test whether there is an enrichment of tissue-specific eQTLs from the GTEx consortium analyses within our genome-wide association study results across infiltration phenotypes (ieQTLs). We first relaxed our significance threshold to the GWAS catalog cut-off by querying all genetic associations with $P < 10^{-5}$ to represent an expanded set of iQTLs (Supplementary Data 11). Then, for each iQTL, we randomly sampled 100 new genome-wide variants that have similar minor allele frequencies and linkage disequilibrium patterns. For each infiltration phenotype, we then used the generated SNPs to test whether there were excess observed tissue-specific ieQTLs compared with random segregation of iQTLs and eQTLs (Methods section). Our testing framework discovered that 23/189 infiltration phenotypes (12.1%) had significant enrichment of tissue-specific eQTLs in the genetic associations ($FDR < 0.1$; Fig. 5i; see Supplementary Note 16). Furthermore, we observed an overall excess of these tissue-specific eQTLs across all infiltration-variant associations (804 observed eQTLs versus 757 expected eQTLs in 12086 iQTLs; $P = 0.04$). We note that directionality is unclear when SNPs are both iQTLs and eQTLs. These SNPs could directly alter both immune content and expression levels (such as through altering transcription factor activity), but it is also likely that gene expression differences drive phenotypic changes in infiltration patterns. Alternatively, it is possible that cellular heterogeneity differences (from infiltration effects) underlie many previous eQTL associations (since differences in the sample's cell-type composition will influence population-level measurements; Supplementary Fig. 20).

To attempt to infer key functions involved in infiltration from our genetic results, we used the tissue-specific gene expression associations from our infiltration-associated iQTL variants (the set of ieQTLs in Supplementary Data 12). We constructed a GeneMania network[49] by forming a list of ieGenes (genes whose expression is significantly associated with the iQTL variant, as determined from tissue-specific GTEx analyses[16]) from ieQTLs with our relaxed iQTL threshold ($P < 10^{-5}$). We queried 179 total genes, building a network of 209 genes with 30 additional relevant genes (Supplementary Fig. 22). We found that this network was significantly enriched for many immune functions, including but

not limited to: antigen processing & presentation, MHC II activity, lymphocyte costimulation, and response to interferon gamma ($P < 10^{-15}$ for each) (Supplementary Data 13). Additionally, the most interconnected added genes were involved in MHC activity and GeneMania identified a 30.52% weighting enrichment between network nodes to MHC-related protein domains collected in InterPro (Supplementary Data 15). In summary, our GeneMania network highlights central mechanisms controlling the immune environment.

**Analysis of iQTL effects across infiltration phenotypes.** Lastly, we analyzed whether iQTLs commonly displayed pleiotropic effects. First, we found no iQTLs or ieGenes associated with multiple infiltration phenotypes under a genome-wide significance threshold ($P < 5 \times 10^{-8}$). Under a relaxed $P < 10^{-5}$ threshold, we still found that almost all iQTLs and ieGenes were associated with only a single cell type in a single tissue type (iQTLs: 99.8% associated with only a single tissue type and 96.9% associated with only a single cell type; ieGenes: 97.2% associated with single tissue type and 94.8% associated with a single cell type; Supplementary Data 14, Supplementary Table 6). In addition, the set of multi-tissue iQTLs and ieGenes were identified in only two total tissue types. While a larger dataset would be better able to assess the effects across multiple phenotypes, our results suggest that genetic variant effects primarily drive a specific immune signature in a particular tissue type. When considered together with the previous observation that there is an enriched signal of tissue-specific eQTLs in the iQTLs, our results begin to suggest that each tissue has its own unique set of genetic influences.

## Discussion
The GTEx consortium project enabled an analysis of transcriptomic variation across diverse human tissues, and the discovery of association between that variation and genetic polymorphisms. With the development of computational algorithms that can deconvolute the cellular heterogeneity underlying bulk RNA-seq data, the GTEx data sets could be utilized to evaluate the baseline immune landscape across the human body. This quantification of immune cells from bulk RNA-seq should become standard in many bulk RNA-seq analyses. Additionally, we observed that population-level gene expression values are strongly affected by the abundance of immune cells through a principal component-based correlation analysis, implying that measures of cellular heterogeneity should be considered in downstream analyses of bulk expression data.

We also developed frameworks to leverage information across multiple cell-type estimation methods and capture differences in deconvolution methods. An individual cell-type estimation method has imperfect correlation with the true scores and other computational methods, likely due to the selected markers and cell types in the reference set inducing certain biases. Thus, we increased our confidence in downstream analyses by incorporating results from multiple deconvolution algorithms, and demonstrated convergence on plausible, significant results while maintaining low false positive rates in large genomic analyses.

We also note that in the clinic, the relative ratio of CD4:CD8+ T cells is a blood test marker to monitor the health of the immune system[50], which can be better captured by relative estimates of immunity. Our demonstration of heterogeneous immune content across tissues suggests that deconvolution methods can potentially be used to derive an expanded set of biomarkers to assess immunologic health across a variety of organs.

Importantly, we demonstrated substantial variability in immune content across individuals within a number of different tissue types. Inspired by efforts to characterize the heterogeneity of tumor immune landscapes, we engaged in an endeavor to discretize clusters of individuals based on their tissue's immune content[51–53]. We were able to successfully show strong separability of immune content in hot clusters compared to cold ones across most phenotypes, but were unable to define tissue-specific differentially expressed genes driving the variability. Nevertheless, we identified genetic loci associated with immune cells that could pose as future clinical markers to guide patient stratification. In cancer, the infiltration profile may be driven by not only new somatic mutations but also pre-existing germline variants. Since the immune signature in the tumor microenvironment is highly correlated with the response to treatments such as immunotherapy, germline variants could enhance predictive modeling of response and reveal novel therapeutic targets for shifting infiltration profiles to a more favorable one. Previous studies demonstrated that cancer cells maintain chromatin structure from the tissue-of-origin, so it is possible that germline iQTLs have conserved infiltration effects in the cancer cells[54]. If this were the case, then functional experiments could be a promising avenue for developing medicines to shift infiltration patterns. Overall, understanding a personalized baseline immune response from genetics would enhance the interpretation of immune presence in the tumor microenvironment.

An important area of future research is to test associations between somatic mutation burden and immune cell estimates in healthy tissues. Recent research performed using the GTEx database has shown that genetically distinct non-cancerous subclonal populations may arise in healthy tissues, akin to the phenomenon of clonal hematopoiesis of indeterminate potential in blood cells[55]. We saw a significant increase of T cell content in female breast and thyroid tissues compared to males. Immune response may correlate with somatic mutation detection, and it would be interesting to evaluate whether differential immune content between males and females is driven by sex-differential somatic clonal mutations which could eventually promote cancer. In parallel, the epidemiological differences between males and females in developing breast and thyroid cancer is striking: of breast cancer, there are 100 times more cases in females compared to males, and of thyroid cancer, there are nearly three times as many cases in females compared to males[25]. Further investigation could provide improved understanding of the increased female disease incidence.

Lastly, we note that superior computational algorithms for cell-type estimation are still needed, as well as larger and better annotated data sets. The algorithms we used are limited to inference of cell-type compositional information and do not infer a cell's molecular signatures, where dysregulation may be even more informative[56]. Furthermore, the available reference profiles do not allow differentiation between tissue-resident and tissue-infiltrating cellular subsets, which would require custom reference profiles based on single-cell RNA-seq (scRNA-seq) data[57]. While scRNA-seq information can provide a more intricate perspective of the infiltrating immune cells, the true cell-type proportions in the samples are distorted from tissue dissociation during the sample preparation process[57]. Therefore, single-cell sequencing may potentially be inferior to deconvolution for enumerating cell-type fractions from solid tissue biopsies. Furthermore, single-cell studies have not been scaled large enough to understand the genetic basis of infiltration patterns. Even with bulk sequencing, the sample sizes examined in our study are limited and must be expanded. We limited our statistical genetic analysis to tissues having greater than 70 samples with matched genotype and phenotype information, and the maximum tissue type had 361 samples. At this sample size, we can only detect the largest of genetic effects. If infiltration is a widely polygenic trait, then increased sample size is necessary to dissect the genetic

architecture of inter-individual differences in infiltration. Similarly, larger sample sizes will allow potential detection of tissue-specific immunomodulatory genes in our hot-cold analysis. Finally, it would enable improved assessment of infiltration pleiotropy. In our study, our identified genetic variants were rarely associated with multiple immune infiltration phenotypes. This implies that the genetics of infiltration differs depending on the tissue of interest and the expression patterns in that tissue, and that down-regulation of one tissue's key functional genes within another tissue could create a completely separate genetic variation network that leads to infiltration. However, a larger dataset is necessary to ascertain how tissue-specific the genomics of infiltration patterns are.

## Methods

**GTEx data.** Processed gene expression profiles from the GTEx v7 data release were downloaded from the GTEx data portal and used for deconvolution and expression-based analyses. Genotype data and raw fastq reads are from the GTEx v6 release and were downloaded from dbGAP. The GTEx v6 data were used in all other analyses. We focused our analysis on 46 of the 53 tissue types with $N > 70$ and not derived from a cell line.

**Simulating immune-spiked synthetic mixes.** To generate "immune-spiked" synthetic mixes, we hand-selected one sigmoid colon GTEx sample (GTEX-XXEK-1826-SM-4BRVC) and one sun-exposed skin GTEx sample (GTEX-WFON-2126-SM-3LK7O) from the v6 release. Both these samples were identified by applying CIBERSORT-Absolute to all GTEx samples and identifying samples with the lowest detected presence of infiltrated immune cells (high CIBERSORT P-values, low cell scores). Using five different CD4+ T cell references and five different CD8+ T cell references (Supplementary Data 16 for SRA), we designed 90 synthetic mixes which contained 80–95% reads sampled from one of the GTEx samples and 5–20% of the reads sampled from the T cell samples. There were four different simulation types: (1) only CD4+ T cells infiltration as 5–20% of the sample, (2) only CD8+ T cells infiltration as 5–20% of the sample, (3) both CD4+ and CD8+ T cells infiltration in equal proportions as 5–20% of the sample, and (4) CD4+ and CD8+ T cells infiltration but in unequal proportions (2:3 and 1:4 ratios as 5–20% of the sample). Half the simulations were created using 1 CD4+/CD8+ reference and half with 5 CD4+/CD8+ references (cellular heterogeneity versus no heterogeneity). Both the colon and skin samples represented half the simulations, and the skin and colon samples were not part of any of the same synthetic mixes. The different immune samples used and their relative proportions across the 90 synthetic mixtures generated is outlined in Supplementary Data 16.

The bulk tissue and immune samples were aligned to the GRCh38 reference genome using STAR[58] and sorted with samtools[59]. The number of reads in each sample were quantified using samtools idxstats, then downsampled to the desired library size using samtools view with the -s flag and the specified percentage of total reads. Next, the resulting bam files containing the downsampled bulk and immune reads were merged using bamtools merge to create a single synthetic mixture bam file[60].

**Generating TPM gene measurements from the synthetic mixes.** RNAseq samples were quantified with the Gencode gene annotation reference (V22 release). Aligned reads were then quantified for gene expression in terms of TPM and FPKM using StringTie[61].

**Deconvolution of bulk RNA-seq profiles.** To deconvolute bulk RNA-seq profiles into cell type scores, we used CIBERSORT-Relative, CIBERSORT-Absolute, and xCell. CIBERSORT P-values were generated using 1000 permutations and quantile normalization disabled. When estimating immune content using xCell, the scores were generated separately for each tissue type. Since the scores are estimated by assessing the variability across all input samples, the inclusion of distinct tissue types can be a confounder to the estimation. We demonstrated this by testing on the synthetic mixes and showing that xCell scores are worse when the algorithm is applied to multiple tissues simultaneously, compared to one tissue at a time (see Supplementary Table 2).

**Empirical evaluation of deconvolution methods.** To empirically compare relative and absolute scores, we used our synthetic mixes. We calculated the true amount of infiltration as two separate measures: "tissue" and "immune cell". In the former, true amount of infiltration is calculated as the percent of reads from the immune cell type in the entire sample. In the latter, the true amount of infiltration is calculated as the percent of reads from the immune cell type in the immune content of the sample.

We deconvoluted each synthetic mix transcriptome into three different cell-type scores each for CD4+ T cells and CD8+ T cells, using CIBERSORT-Relative, CIBERSORT-Absolute, and xCell. We describe the derivation of these cell-type

scores below. All three estimates, regardless of generation process, were correlated with the true amount of infiltration in both the "tissue" and "immune cell" scenarios to quantitatively assess the differences.

**Merging cell subtype estimates into single scores.** From the default deconvolution scoring by CIBERSORT-Relative, CIBERSORT-Absolute, and xCell, we defined 14 specific immune cell type phenotypes (CD8+ T cells, CD4+ naive T cells, CD4+ memory T cells, helper T cells, regulatory T cells, gamma delta T cells, B cells, NK cells, neutrophils, macrophages, dendritic cells, mast cells, monocytes, and eosinophils) and 2 additional broader immune cell type phenotypes (lymphoid-based and myeloid-based cells), for a total of 16 cell types. The scores for these phenotypes are calculated by taking the sum of multiple original default cell-type scores, which we have defined in Supplementary Tables 3 and 4. The 14 distinct immune cell types combine the scores from unique cell types in the reference such that there is no overlap. (Note that there is an overlap between the broader lymphocyte/myeloid cell scores with the 14 distinct immune cell types.)

We also calculate a CD4+ T cell score, which is only used in our simulation testing and to calculate CD4:CD8 T cell ratios in whole blood within our supplementary analysis. The CD4+ T cell scores are calculated as the sum of "CD4+ naive T-cells", "CD4+ Tcm", "CD4+ Tem", "CD4+ memory T-cells", "Th1 cells", "Th2 cells", "Tregs" in xCell, and the sum of "T cells CD4 naïve", "T cells CD4 memory activated", "T cells CD4 memory resting", "T cells follicular helper", "T cells regulatory (Tregs)" in CIBERSORT.

**Analysis of immune content across GTEx tissues.** Within each tissue, the median value for each of 14 cell types was calculated separately for each deconvolution method. The maximum values for xCell, CIBERSORT-Absolute, and CIBERSORT-Relative median scores were set to 0.05, 0.6, and 0.3 respectively to enhance the visualization of variability across tissues and cell types and minimize outlier influence. Separately, for each deconvolution method, hierarchical clustering of tissues was performed on the 14 distinct cell types' median scores using Euclidean distance (excluding lymphocytes and myeloid cells). The clustering results were visualized using *pheatmap*[62] under the "complete" linkage setting, to analyze similarity between tissue immune content.

Separately for each deconvolution method, we calculated pairwise correlations between all infiltration phenotypes. We plot these correlations in heatmaps, and examined whether correlations between infiltration phenotypes is conserved across the three deconvolution methods (Supplementary Fig. 5).

We visualized a single cell type across all tissues for each deconvolution method using boxplots, sorted by the CIBERSORT-Absolute cell type score (Supplementary Figs. 6–9).

Lastly, we used t-SNE to visualize immune content within a single tissue type and identify whether any clusters exist. We used scatterplots to visualize the two components and colored each point (which represents a unique sample or individual) by measured CD8+ T cell content (Supplementary Fig. 10).

**Defining infiltration phenotypes and filtering for analysis.** Infiltration phenotypes are a tissue-by-immune cell type pair; they are defined by the estimated amount of a particular immune cell type measured in a specific tissue on a continuous scale. As described previously, each infiltration phenotype included three separate measurements (by CIBERSORT-Relative, CIBERSORT-Absolute, and xCell). By measuring 16 cell types in 46 GTEx tissues, which represent the 14 specific immune cell types and the 2 broader immune cell types described previously, we generated 736 original infiltration phenotypes. We then reduced to 189 more-informative phenotypes by using the filtering criteria described below:

(1) The tissue has sample size of $N > 70$. This only includes samples with matched genetic, expression, and covariate information. (This is also a threshold used by the GTEx consortium's eQTL analysis.)

(2) The tissue often has detectable immune cells in the samples, as measured that > 50% of samples from that tissue have CIBERSORT-Relative P-values < 0.50. (Null hypothesis: no immune cells from the LM22 reference are present in the sample.)[19]

(3) The immune cell type is, on average, a substantial part of the immune content measured in that tissue, as measured by >5% mean score across CIBERSORT-Relative deconvolutions and >0.001 mean score across xCell estimations.

(4) CIBERSORT-Absolute and xCell scores have non-negative correlation. (This prevents measurements where the deconvolutions inversely correlate and actually disagree.)

(5) The GTEx tissue type is not a cell line (thus, EBV-transformed lymphocytes and fibroblasts were removed).

**Analyzing principal components of gene expression profiles.** Principal component analysis was performed on the processed gene expression matrix for each tissue separately. A linear regression model was fit between the CIBERSORT-Absolute scores for each infiltration phenotype and, one-by-one, each of the first four principal components in that tissue. The P-values across all results (756 tests)

were adjusted using Benjamini & Hochberg's false discovery rate (FDR) correction[21]. To assess significance, we tested at $FDR < 0.1$.

**Characterization of consensus clusters.** For each infiltration phenotype, we performed Euclidean distance-based k-means consensus clustering on the $1 \times N$ vector of deconvolution scores for a particular deconvolution output (e.g. xCell). To implement this, the $1 \times N$ vector was cloned into an $3 \times N$ matrix of three identical rows, which is required as input into the consensus clustering algorithm implemented in the BioConductor package ConsensusClusterPlus[63]. Clustering was sped up using the *fastcluster* R package[64] and 2000 resampling cycles were performed. The algorithm was set to identify a minimum of 2 clusters and a maximum of 20 clusters. We then chose the number of clusters based on the maximum observed relative change in area of the empirical cumulative distribution function, which is a measure of robustness:

$$\mathrm{CDF}(c) = \frac{\sum_{i<j} 1\{M(i,j) \le c\}}{N(N-1)/2} \qquad (1)$$

Here, 1 denotes the indicator function, $M(i, j)$ denotes entry $(i, j)$ of the consensus matrix $M$, and $N$ is the number of rows (and columns) of $M$. The Consensus clustering algorithm returns the consensus cluster assignments for each sample based on the number of clusters chosen and the 2000 resampling cycles of that cluster estimate.

For a given infiltration phenotype (e.g. macrophages in lung tissue) and deconvolution measurement (e.g. xCell), the mean cell type score (e.g. macrophage score) was calculated for each cluster. All samples in the cluster with the maximum score were assigned a label of "hot" and all samples in the cluster with the minimum score were assigned a label of "cold". All other samples were labeled "intermediate". This procedure was repeated separately for each of the xCell, CIBERSORT-Absolute, and CIBERSORT-Relative scores from the 189 infiltration phenotypes (for a total of 567 runs). Finally, samples consistently identified as hot across all three sets for a given phenotype were taken as "consensus" hot samples. The same was repeated for labeling consensus cold samples. This final set of hot and cold samples were considered for differential expression analysis (see Fig. 3a for graphical methods; Supplementary Figs. 11 and 12).

**Characterization of quintile clusters.** For each infiltration phenotype, the hot cluster represented the individuals that had the highest 20% scores in each of the three deconvolution outputs (xCell, CIBERSORT-Absolute, and CIBERSORT-Relative). The cold clusters represented the individuals that had the lowest 20% scores in each of the three deconvolution outputs.

**Differential expression analysis of infiltration patterns.** Differential gene expression was performed between the consensus hot and cold samples for each tissue-cell pair using limma-voom[65]. We required that there be at least 6 hot and 6 cold samples in each infiltration phenotype before proceeding with differential expression analysis. This helped address problems related to a lack of sufficient sample size in hot and cold clusters.

In our differential expression model, we used age (numeric; binned into 10-year categories), sex (binary), death classification (categorical; 0, 1, 2, 3, 4), autolysis score (numeric), and sample collection site (categorical) as covariates. Categorical covariates were only included if there were a minimum of three hot and three cold samples in each level of that covariate. However, if there was a single level of a covariate that did not feature hot samples, we required that there be <5 cold samples in that level for the covariate to be included (Supplementary Data 4).

We applied strict statistical thresholds to narrow down the large number of differentially expressed genes for pathway analysis: $FDR < 0.01$ and log fold-change $\geq 2.0$, after adjustment for covariates (Supplementary Data 5). Lastly, we aimed to identify tissue-specific pathways from our differentially expressed genes by using Ingenuity Pathway Analysis (IPA) software (Supplementary Data 6–8). All phenotypes with at least 5 differentially expressed genes were used as input into IPA. We analyzed the shared results across tissues by identifying common pathways, genes, and transcriptional regulators from our results. This was repeated using a pre-filtered and post-filtered list, where genes that were by the CIBERSORT or xCell reference were removed.

**Tissue-specificity of infiltration patterns.** We explored whether individuals hot in one tissue type were more likely to be hot in other tissue types. For each cell type, all individuals with at least eight tissue samples represented within the infiltration phenotypes (for that cell type) were identified. The median and mode number of hot tissues within these individuals were calculated. Hierarchical clustering was performed between tissues and individuals, where binary values represent hot or not in a particular tissue for each individual.

To formally analyze whether hot patterns in one tissue are independent of hot patterns in other tissues, the immune-hot clusters from the infiltration phenotypes were assessed using a Fisher exact test. This was performed as follows. First, for a particular cell type, all tissues used within the 189 infiltration phenotypes were identified. Next, for each possible pair of these tissues, all individuals who contributed samples to both tissue types were identified. A two-by-two contingency table was then created for each tissue pair, where samples are classified as hot or

not hot in each tissue. Finally, a Fisher exact test was used to assess the null hypothesis that the two tissues exhibited independent hot-sharing patterns. Non-independent hot-sharing patterns indicates that the probability of one tissue being inflamed is conditional on another tissue being inflamed. This process was repeated across tissue pairs for all cell types, and a Bonferroni correction was used to assess significance. The procedure was performed on hot clusters based on top 20% scores across all three deconvolution methods (quintiles), top 40% scores across all three deconvolution methods (top two quintiles), or consensus k-means clustering (described previously in greater detail).

**Analysis of age and sex effects over tissue immune content.** A multiple linear regression model accounting for age (numerical; discrete, binned into 10-year categories), sex (binary), death classification (categorical; 0, 1, 2, 3, 4), autolysis score (numerical), and sample collection site (categorical) covariates was fit for each phenotype to estimate age and sex effects. This was repeated for each of the deconvolution methods, and the *P*-values were combined using Empirical Brown's method[22]. This method uses a covariance matrix to combine dependent *P*-values, allowing the incorporation of distinct analyses from each deconvolution method. To calculate combined *P*-values for a specific infiltration phenotype, a $3 \times 3$ covariance matrix was calculated from the three deconvolution outputs which describe the phenotype. (Therefore, a separate covariance matrix was calculated for each infiltration phenotype.) As a final step, Benjamini & Hochberg's false discovery rate (FDR) correction[21] was applied to adjust all age-based *P*-values, then to separately adjust all sex-based *P*-values. Significance was assessed at FDR < 0.1. Empirical Brown's method *P*-values are reported in the text. Lastly, we visualized differences in breast tissue heterogeneity by applying t-distributed stochastic neighbor embedding (t-SNE)[66] to the full (original) 64-cell type infiltration matrix from xCell. Venn diagrams were plot using the *limma* package.

Myeloid:lymphoid and CD4:CD8 ratios were also calculated in whole blood. To avoid infinite ratios and extreme outliers due to lymphoid or CD8+ T cell counts being equal to zero, two steps were performed when calculating ratios. First, $\varepsilon = 10^{-10}$ was added to both the numerator and the denominator prior to calculating the CD4:CD8 and myeloid:lymphoid ratios. Second, to avoid extremely large outliers, a rank-inverse normal transformation was applied to the phenotype. Age and sex were tested for association with these ratio phenotypes using an analysis procedure nearly identical to the one described above. The difference is that only two deconvolution outputs were used for analysis, xCell and CIBERSORT, and therefore 2×2 covariance matrices were calculated. This is because CIBERSORT-Relative and CIBERSORT-Absolute scores would return identical values for any ratio values. To plot myeloid:lymphoid ratios in Supplementary Fig. 17, a linear model containing all previous covariates, except for age, was fit to the ratio phenotype. Model residuals were calculated and used for plotting.

**Pre-GWAS: genotype and phenotype processing.** To identify genetic variants associated with infiltration phenotypes, we first removed covariate effects by adjusting phenotypes using a multiple regression model. This model was used with the following covariates: age (numerical; discrete, binned into 10-year categories), sex (binary), death classification (categorical; 0, 1, 2, 3, 4), autolysis score (numerical), sample collection site (categorical), and three genotype-based principal components (to control for any population stratification). Genotype-based principal component analysis was performed using the –pca function in *plink*[67] across all GTEx v6 individuals. We note that gene expression-based latent factors, such as PEER factors[68], have been demonstrated to be a powerful approach to correct for unwanted noise and technical variation. However, as described previously, our gene expression-based principal components correlated strongly with deconvolution estimates. As a result, our gene expression-based principal components, which could drastically reduce statistical power and inflate false positive rates, were not included in the model. We used this model to calculate residuals. Residuals were transformed into *z*-scores using a rank-inverse normal transformation. Transformed residuals were then used for GWAS testing. Genetic SNPs were filtered based on minor allele frequency (<0.05), missingness (>0.1), and Hardy–Weinberg Equilibrium *P*-values ($<10^{-6}$). A total of 5.6 million SNPs remained for analysis.

**Performing the genome-wide association study.** We tested for any associations between genome-wide variants and infiltration phenotypes using simple linear regression implemented in *PLINK* (plink –assoc). This analysis returned three *P*-values for each SNP, one for each deconvolution method (xCell, CIBERSORT-Relative, and CIBERSORT-Absolute), which were combined using Empirical Brown's method into a single *P*-value. In all, $3 \times 3$ covariance matrices for Empirical Brown's method were calculated based on the transformed residual phenotypes from the covariate analysis described previously. All SNPs below a genome-wide threshold of $5.0 \times 10^{-8}$ were considered "genome-wide significant", while a threshold of $P < 1.0 \times 10^{-5}$ was used for "suggested significance". This less stringent threshold was used in downstream system-wide analysis.

**Identifying other associations with iQTLs.** The Michigan Genomics Initiative PheWeb[38] was used to search for association between rs6482199 and thyroiditis.

The GWAS results for self-reported thyroiditis in UK Biobank were downloaded from the Neale lab analysis[39] to identify rs6482199's association. We used phenoscanner[69] to identify any other associations with rs6482199 variants. Next, we used GeneHancer[43] to query all promoters or enhancers linked to both COMMD3 and DNAJC1 and identify the genomic regions. We queried 49 UK Biobank SNPs in these regulatory regions with $MAF > 0.01$. We next used the Neale lab analysis of UK Biobank to test whether any of these SNPs were associated with self-reported thyroiditis. We corrected for significance using a stringent Bonferroni correction of $P < 0.05/49$ SNPs.

For the rs9989443 and rs56234965 variants, we identified other phenotypes that have been previously associated by using phenoscanner. We input the gene start and gene end positions to query all GWAS-SNP associations for variants within DRICH1 and CCDC40.

We also tested for enrichment of previous GWAS hits within the 31 genome-wide significant iQTLs. We used phenoscanner to identify whether an iQTL has been previously associated with a GWAS phenotype ($P < 5 \times 10^{-8}$). Then, we randomly sampled 31 SNPs from the genome for ten iterations. In each iteration, we calculated the proportion of the 31 SNPs that has been associated with a GWAS phenotype ($P < 5 \times 10^{-8}$). We calculated the mean proportion across these 10 iterations. Finally, we performed a one-sided binomial test using the mean proportion, the observed number of GWAS-iQTLs, and the observed number of iQTLs as parameters to calculate enrichment significance.

**Gene expression versus infiltration phenotype**. The relationship between raw gene expression and infiltration phenotypes were tested using Pearson's correlation test (cor.test() function in R). Test P-values were reported.

**Genetic network analysis using GeneMania**. GeneMania combines multiple biological databases with a weighted "guilt-by-association" algorithm to add relevant genes to the query list and identify network edges[49]. Suggested ieQTLs (loci associated with an infiltration phenotype that are also GTEx single-tissue eQTLs in that tissue; $P < 10^{-5}$) were used to form a list of ieGenes (the target genes of ieQTLs). The list of genes (recorded in Supplementary Data 11) were uploaded to the GeneMania software to construct a network of the input genes. In this analysis, up to 15 relevant functional attributes and 30 genes were allowed to be supplemented to an original query of 179 ieGenes. ieGenes with no shared edges with any other ieGenes were removed. To quantitatively assess the connectivity of each newly added gene to the network, GeneMania computes a score which was used to rank and identify the most interconnected genes.

**Testing for eQTL enrichment in iQTLs across phenotypes**. iQTLs ($P < 10^{-5}$; listed in Supplementary Data 12) were tested for an enriched overlap with eQTLs. For each phenotype, we generated the list of N iQTLs and match each of the N variants with a list of similar variants, as determined by minor allele frequency (within 1%, as calculated using –freq in plink from all GTEx individuals' genetic data) and the same number of variants in linkage disequilibrium (LD) ($r^2 > 0.2$, as calculated in the 1000 Genomes Phase I EUR genetic data[70] and downloaded from Haploreg v4[42]). (We note that the threshold requiring the number of variants in LD to be identical is relaxed to plus-minus 1 variants-in-LD when no such variants exist.) We use these lists to generate 100 permutations. For each permutation, we randomly sampled 1 matched SNP for each of the N iQTLs. From the list of N randomly sampled SNPs, we calculated the proportion of SNPs that are same-tissue GTEx eQTLs. From these 100 permutations, we calculated 100 eQTL proportion measurements. We then calculated the mean proportion, which we refer to as q. We let x be the # iQTLs that are eQTLs in that tissue. Lastly, we performed a one-sided binomial test with x equal to the number of successes, N equal to the number of trials, and q equal to the hypothesized probability of success. We tested the null hypothesis that the observed ieQTL proportion is significantly different than random sampling. This approach is summarized in Supplementary Fig. 19. We also implemented a two-sided version of the test, with results summarized in Supplementary Fig. 21.

We also calculate the expected number of ieQTLs across all infiltration phenotypes and compare to the observed number of ieQTLs in iQTLs. We do this by computing a phenotype-wide q by weighting each phenotype by N. Therefore, $q_{new}$ is equal to $(N \times q)/N$, and the expected number of ieQTLs is $q_{new} \times N$. Then, a one-sided binomial test was used to assess significance.

**Reporting summary**. Further information on research design is available in the Nature Research Reporting Summary linked to this article.

## Data availability

The genetic data that supports the findings of this study can be found under dbGaP study accession phs000424.v8.p2 as the v6 release. The gene expression information can be found using the v7 release from gtexportal.org. All other data are included in the supplemental information or available from the authors upon reasonable requests. The source data underlying Figs. 1a, 2a, b, 3b, 4a–g, and 5a, b, e–i are provided as a Source Data file.

## Code availability

All computer code used to support the findings in the manuscript are located at the following GitHub repository: https://github.com/drewmard/GTEx_infil.

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

## Acknowledgements

We would like to thank members of the Clark laboratory and members of the Elemento laboratory for discussion surrounding this project and Zakieh Tayyebi for help designing a body-infiltration map (Fig. 1a). We thank the GTEx donors for their contributions to science and the GTEx consortium for generating raw and analyzed data for researchers to access. O.E. is supported by Janssen and Eli Lilly research grants, NIH grants UL1TR002384, R01CA194547, and LLS SCOR grants 180078-02, 7021-20. Support was also provided for A.R.M. and A.V. by the Tri-Institutional Training Program in Computational Biology and Medicine.

## Author contributions

A.R.M., A.G.C., and O.E. conceived and designed the study. A.R.M and M.U. performed the GTEx analysis. A.R.M., M.U., and A.V. designed and performed the simulations to test the deconvolution methods. A.V. and B.B. aided in developing the analysis methods and provided samples for the simulations. Z.T. led development of the body-infiltration map in Fig. 1a. J.M. provided access to GTEx data. A.R.M., M.U., A.G.C., and O.E. wrote the manuscript. A.G.C. and O.E. supervised the study. All the authors reviewed and approved the manuscript.

## Competing interests

O.E. is scientific advisor and equity holder in Freenome, Owkin, Volastra Therapeutics and One Three Biotech.
