## [Peer Review File · Nature Communications]

Reviewers' comments:

Reviewer #1 (Remarks to the Author):

In this manuscript, Marderstein and colleagues analyze immune composition in 53 healthy human tissue types (GTEx) and characterize how tissue-specific immune content varies as a function of age, sex, and genetic polymorphisms. To do this, the authors first establish a computational pipeline for deconvolving bulk tissue RNA-seq data using two commonly applied tools, CIBERSORT (relative and absolute modes) and xCell. They then (1) analyze global immune abundance patterns in GTEx, (2) identify and explore differentially expressed genes (DEGs) between tissues with inflamed and non-inflamed transcriptional states, and (3) determine and characterize correlations between 73 immune infiltration phenotypes and age, sex, and quantitative trait loci (QTL). The authors uncover a number of novel associations between estimates of infiltrating immune cell levels in specific tissue types and the presence of genetic polymorphisms, including QTLs previously associated with gene expression by the GTEx consortium.

As the authors emphasize, new insights into tissue-specific immune cells could advance our understanding of immunity in human disease, including cancer. Unfortunately, the current manuscript falls short of achieving this goal on several levels. For example, the authors miss an opportunity to link their findings to human disease - do any of the identified iQTL or ieQTL associations hold in cancer, auto-immune disorders, or other pathologies? Do patterns of normal tissue-specific immune content, including inflamed and non-inflamed dichotomies, correlate with frequencies of tumor-specific immune content in cancer? These questions should be addressable using existing datasets. Additionally, key methodological details are poorly described. Collectively, these issues significantly dampen enthusiasm for the current work.

Major comments:

1. As alluded to above, the authors motivate their study by linking it to human disease. In particular, they emphasize the need to better delineate the biological underpinnings of hot vs. cold tumors as a means of improving response to immunotherapy. Yet no direct evidence is provided to link the findings in this paper to immune infiltration patterns in human tumors. The authors should attempt to address this issue (e.g., in TCGA), as otherwise the manuscript is completely descriptive and its potential relevance to human disease is unclear.
2. The authors explore differences between relative and absolute measures of deconvolved immune content. While strong arguments can be made in favor of both measures, they each quantify immune composition in fundamentally different ways (e.g., Supp. Figs 6, 7, and 9). This raises the question of whether the authors' decision to consider relative and absolute measures in a combined model (by Empirical Brown's method) is sensible and whether different conclusions might be obtained by evaluating each measure separately.
3. Although grouping immune cell types into major lineages is understandable, the authors omit several key tissue-associated immune cell types from their correlative analysis without explanation (B cells, NK cells, dendritic cells, mast cells). This unexpected omission needs to be addressed as the current work is incomplete without them.
4. Age-related changes in the immune system have been previously described, particularly in the bone marrow and blood. For example, hematopoietic stem cells become myeloid-based with age (PMID 22123971). As a control, did the authors find evidence for myeloid-lymphoid skewing in aged blood by their deconvolution pipeline (e.g., as seen here in Fig 2 in PMID 26808160)?
5. Given substantial differences in mammary tissue biology between males and females, the identification of sex-specific differences in immune content in breast tissue is not surprising. A tissue type with less obvious differences in fundamental biology would be preferred.
6. In the analysis of DEGs between "hot" and "cold" tissues, the authors identify a preponderance of

immune-related genes, as expected. An analysis that controls for immune cell composition via a linear model would better capture expression differences that are not a surrogate for total immune cell content.

7. The infiltration phenotypes used extensively throughout this work are very unclear and their derivation is poorly described. This makes it difficult to assess the analyses and conclusions that rely on these signatures.

8. Similarly, the approach used to delineate inflamed and non-inflamed tissues is not clearly described or graphically depicted, and the panel in figure 1d is difficult to understand. How was the infiltration z-score calculated?

Minor comments:

1. The authors conflate tissue-resident and tissue-infiltrating immune subsets. Macrophages are often tissue-resident as are resident memory T cells, mast cells, etc. While this is primarily a semantic issue in the current work, the authors should address this issue with more nuanced language.

2. An analysis of the p-values obtained by all three deconvolution techniques would be helpful to understand their concordance and how the covariance structure is being exploited by Empirical Brown's method to produce meta-p-values.

Reviewer #2 (Remarks to the Author):

In this manuscript, Marderstein and colleagues present a retrospective study of the GTEx collection, aiming to infer determinants (genetic or otherwise) of immune infiltration. While the gene expression is provided on a tissue level, the authors utilize deconvolution tools to estimate the cellular composition that underlies the (observed) tissue average. With these estimations in hand, the authors quantify the extent of immune infiltration to various tissues and then turn to explore the relation between these estimated phenotypes and covariates of interest, and identify for instance, a significant association between the strength of neutrophil abundance in the lung and genetic variant that reside near a key transcriptional regulator. Overall, this is an important contribution, which builds on, and to some extent helps realize the potential of community efforts such as GTEx. Having said that, there are a number of caveats in the methods and in the interpretation of the results, which we describe in more detail below.

1. Stratification and analysis of "hot" vs. "cold" samples:

a. The procedure for labeling samples as "hot" or "cold" is not clearly described; please revise the respective methods section and provide a clearer (yet short) explanation in the main text. It is my understanding is that for a given deconvolution algorithm the procedure labels *all* samples as either hot or cold. It therefore runs the risk of making arbitrary decisions when the signal is not conclusive either way (notably, this effect may be mitigated by the need to have consistent assignment across the three algorithms). How stable is this procedure? (e.g., to sub-sampling). How will it change the results (e.g., comment 1b below or on the lack of consistency in inflammation patterns across tissues) if it focuses on the "extreme phenotype" samples and excludes all others?

b. It is a bit unexpected that for ~40% (30 out of 73) of the phenotypes the differential expression analysis does not yield significant outcome, at least out of the very same sets of genes used to stratify the samples. Please provide more detail for why this test (which can be thought of as a sanity check) fails in some cases, as it might mean that the deconvolution and the resulting stratification might be overwhelmed by noise. How correlated are the genes in each of those 30 phenotypes? Are these phenotypes discussed later on in the paper? (e.g., for association with age) and if so, would that lack of DE genes question their validity?

c. "Therefore, we reflected that infiltration patterns are likely tissue-specific, rather than widespread. " This statement may seem to be somewhat contradictory to the results in Figure 1B. Please elaborate.

d. It is not clear from the outset why deconvolution is needed here. Would the DE genes and pathways that came up in this analysis (or other genes and pathways that make sense) be detected in a simpler correlation analysis? (i.e., correlate each gene in the transcriptome with the aggregate expression level of the genes in the same sets used for deconvolution)

e. Since the mode of number of "hot" tissues per individual x cell type was one, it is concluded that "we reflected that infiltration patterns are likely tissue-specific, rather than widespread. " This is quite a strong statement that would benefit from more context and references.

f. While the mode in the latter test is at 1 there is still a non-negligible amount of cases with more than one hot tissue. To support the validity of the results in this part of the paper (i.e., point 1e above as well as the DE analysis [point 1b]) it would be helpful to repeat the deconvolutions with random sets of signature genes per cell type (ideally, matched by mean or median expression) and compare to the resulting numbers of DE genes, numbers of hot tissues per individual, and co-clustering of similar tissues (as in Figure 1b) .

2. Association studies

a. Can the two significant associations be discovered with a more standard analysis, without the need for deconvolution? (namely, associate each variant with the aggregate expression level of the genes in the same sets used for deconvolution).

b. An intriguing question regarding CUX1 is whether its potential effect may be intrinsic to neutrophils or whether it relates to other cellular subsets that are in turn associated with neutrophil abundance. While this is difficult to answer, an additional discussion would be interesting. For instance, how is this gene expressed across the primary cell types in the lung? How does the chromatin look like (in terms of histone modifications, DNA methylation, accessibility in the respective loci) in the respective locus in those cells? While the ideal data needed to address this is unavailable, databases of gene expression and chromatin features per- cell types (such as Immgen) can be interesting to look at. This can further strengthen the support from refs 31 and 32. The same point holds for the second finding (rs116827016)

c. It is not clear whether or not the reported cases of both i- and e- QTL were discovered in the same tissue (e.g., was the eQTL for rs11883564 computed with only sun exposed skin tissues?). If this is not the case, this analysis should be somewhat at odds with the observation that "infiltration patterns are likely tissue-specific, rather than widespread. "

d. To provide further support for the validity of these associations, it would be helpful to repeat the analysis using random sets of signature genes per cell type, as in comment 1f above.

Minor comments

1. Glastonbury et al: this paper has been published. Please update the reference.

2. "Therefore, these results indicate that each method provides interesting information to be exploited in downstream analysis." Not clear how this conclusion is derived.

3. The meaning of "CIBERSORT-Absolute" vs. "CIBERSORT-Relative" as described in the beginning of the main text is unclear and requires further reading and deciphering. Please state explicitly what these actually mean.

4. Figure 3 – data is colored by whether eQTLs are over- represented (red) or under-represented (blue) in the iQTLs. The terminology is a bit misleading. Over/ under- represented implies that there is statistical significance, which means that it can be (and probably often) the case that neither holds.

We thank the reviewers for reading our manuscript and providing their feedback. Both reviewers raised interesting questions and offered intriguing suggestions that we have incorporated into our research. In response, we have made substantial revisions to our manuscript and responded to each of their points. To summarize, we have expanded our analysis from 73 infiltration phenotypes to 189 infiltration phenotypes, which include additional immune cell types such as monocytes, NK cells, and B cells and broader phenotypes for lymphocytes and myeloid-based cells. We have revised our results sections to describe the expanded analysis that has led to new results. Some of the key additional findings include new significant SNP associations, linking these new findings to related disease phenotypes, and an enrichment of MHC activity in gene network analysis during a functional follow-up. We have also answered questions regarding our methods, which has included a modified methods section and all our source code now on GitHub. Please share our GitHub repository containing the source code (https://github.com/drewmard/GTEx_infil) with the reviewers. In our manuscript, **red** text indicates our changes. Below, **purple** text indicates our direct responses to reviewer questions.

Reviewers' comments:

Reviewer #1 (Remarks to the Author):

In this manuscript, Marderstein and colleagues analyze immune composition in 53 healthy human tissue types (GTEx) and characterize how tissue-specific immune content varies as a function of age, sex, and genetic polymorphisms. To do this, the authors first establish a computational pipeline for deconvolving bulk tissue RNA-seq data using two commonly applied tools, CIBERSORT (relative and absolute modes) and xCell. They then (1) analyze global immune abundance patterns in GTEx, (2) identify and explore differentially expressed genes (DEGs) between tissues with inflamed and non-inflamed transcriptional states, and (3) determine and characterize correlations between 73 immune infiltration phenotypes and age, sex, and quantitative trait loci (QTL). The authors uncover a number of novel associations between estimates of infiltrating immune cell levels in specific tissue types and the presence of genetic polymorphisms, including QTLs previously associated with gene expression by the GTEx consortium.

As the authors emphasize, new insights into tissue-specific immune cells could advance our understanding of immunity in human disease, including cancer. Unfortunately, the current manuscript falls short of achieving this goal on several levels. For example, the authors miss an opportunity to link their findings to human disease - do any of the identified iQTL or ieQTL associations hold in cancer, auto-immune disorders, or other pathologies? Do patterns of normal tissue-specific immune content, including inflamed and non-inflamed dichotomies, correlate with frequencies of tumor-specific immune content in cancer? These questions should be addressable using existing datasets. Additionally, key methodological details are poorly described. Collectively, these issues significantly dampen enthusiasm for the current work.

Major comments:

1. As alluded to above, the authors motivate their study by linking it to human disease. In particular, they emphasize the need to better delineate the biological underpinnings of hot vs. cold tumors as a means of improving response to immunotherapy. Yet no direct evidence is provided to link the findings in this paper to immune infiltration patterns in human tumors. The authors should attempt to address this issue (e.g., in TCGA), as otherwise the manuscript is completely descriptive and its potential relevance to human disease is unclear.

As Reviewer #1 mentions, our original manuscript did not link our findings in healthy tissues to potential disease states. Below, we describe two additional analyses we performed to address this issue. First, in our manuscript, we used our most significant associations between a SNP and infiltration phenotype (iQTLs) to identify whether there are any previous phenotype associations for any of the top iQTLs using separate cohorts. Second, in our reviewer response, we show a direct comparison of immune content between healthy GTEx tissues and TCGA cancer tissues.

(1)

First, we used *phenoscanner* to identify whether the top iQTLs from each of the 31 phenotypes with a genome-wide significant association ($P < 5e-8$) have been identified in a previous GWAS ($P < 5e-8$). Using any previous GWAS phenotype, we found that 19.4% of the top 31 iQTLs ($P < 5e-8$) have been associated with another GWAS phenotype in a separate study ($P < 5e-8$). We compared this to random SNPs in the genome, where 5.4% we found were associated with a GWAS phenotype. We used a one-sided binomial test with $q = 0.054$ to test whether the GWAS enrichment in the observed iQTLs were significant, resulting in a p-value = $5.6e-3$.

Next, we linked our two most significant iQTL associations to related phenotypes by analyzing results from already-performed genome-wide association studies in separate, larger cohorts. Below, we summarize our results. A full description can be found in the “Association of genetic variants with infiltrating immune cells” of the revised manuscript.

First, we show that the most significant iQTL, rs6482199, which we found associated with the helper T cell phenotype in thyroid samples, is also associated with thyroiditis in the Michigan Genomics Initiative data and in the UK Biobank data. Furthermore, this variant is associated with *COMMD3* and *DNAJC1* expression in the eQTLGen analysis and overlaps regulatory marks in Roadmap Epigenomics Consortium data. We go on to query six different enhancer and promoter regions that are predicted to be linked to both *COMMD3* and *DNAJC1* and discover another genetic variant in these regions that is significantly associated with thyroiditis in UK Biobank.

We also discuss the 2nd most significant iQTL, the *DRICH1/C22ORF43* intronic variant rs56234965, which was associated with lymphocytes in sigmoid colon samples. We describe how variants located within the start and end positions of *DRICH1* have associated with several intestinal and lymphocytic cancer-related death causes in UK Biobank.

Lastly, we discussed rs9989443 and its association with both *CCDC40* expression and the mast cell phenotype in esophagus (muscularis) tissue. We found that variants located within the start

and end positions of the *CCDC40* gene have been associated with both myeloid leukaemia death and self-reported esophagus disorders in UK Biobank.

All together, this extension to UK Biobank and other data sets allows us to link our genetic findings of the healthy human state to a range of related human diseases. These results suggest that changes in baseline infiltration may alter disease etiology and affect disease risk.

(2)

We downloaded TCGA data and performed CIBERSORT deconvolution of the tumor samples. We found that the scaling factor from CIBERSORT-Absolute was incomparable between GTEx and TCGA (TCGA scaling factor < 1, while GTEx scaling factor is often > 1; this could be due to differences in sample collection and pipeline protocols). As a result, we resort to comparing the two data sets on a purely relative scale (CIBERSORT-Relative). Separately for GTEx and TCGA, we calculated the median CIBERSORT-Relative scores for each of 14 immune cell types in each tissue. We correlated the median cell type scores of GTEx against TCGA.

Overall, we find that there is variability between normal and tumor immune content, but there also exist general similarities (Response Figure 1). Specifically, we see an overall increase in macrophages in TCGA compared to GTEx, and an overall decrease in monocytes and mast cells in TCGA compared to GTEx. However, many T cell subsets have similar quantities in tumor and normal tissue. One clear exception is Tregs, which were rarely identified in GTEx samples but were more often detected in TCGA samples (Response Figure 2). This may be an immunosuppressive mechanism. Tregs can weaken the adaptive immune response and prevent immune recognition, enabling cancer cell growth and tumor progression.

While interesting, we note that we have decided to leave the TCGA analysis out of the main manuscript as we felt that an analysis of tumor infiltration versus baseline healthy infiltration was out of scope from the remainder of the manuscript.

Response Figure 1: Within each of 13 cancers from TCGA and for each of the 14 studied immune cell type phenotypes, the median CIBERSORT-Relative immune cell scores were calculated. The median scores were also calculated in the most similar healthy tissue using GTEx data. The median scores for GTEx and TCGA were plot against each other, where each point represents the median score for one cell type in the TCGA cancer tissue and GTEx healthy tissue. Red line has slope 1, intercept 0. Study abbreviations found here:

<https://gdc.cancer.gov/resources-tcga-users/tcga-code-tables/tcga-study-abbreviations>

Response Figure 2: Similar to Response Figure 1, except each point now reflects a different cancer/tissue type for Treg median scores.

2. The authors explore differences between relative and absolute measures of deconvolved immune content. While strong arguments can be made in favor of both measures, they each quantify immune composition in fundamentally different ways (e.g., Supp. Figs 6, 7, and 9). This raises the question of whether the authors’ decision to consider relative and absolute measures in a combined model (by Empirical Brown’s method) is sensible and whether different conclusions might be obtained by evaluating each measure separately.

We appreciate the concerns by the reviewer on whether to consider relative and absolute measurements simultaneously. As described in our manuscript, relative and absolute measures of deconvolved immune content quantify immune composition in fundamentally different ways. The reviewer correctly states that strong arguments could be made in favor of both measures. We present this in the first section of the paper. However, by using Empirical Brown’s method, we simultaneously perform a data-driven analysis of both perspectives of immune content in solid tissue. We describe these advantages below.

In our manuscript, we show how this combined analysis has increased power over a separate analysis. For example, in our genetic analysis, we describe how the combined Empirical Brown’s method p-values result in a greater number of significant findings (n=31) compared to considering each deconvolution measure (CIBERSORT-Rel, CIBERSORT-Abs, and xCell) separately (n=15, 16, and 16 respectively). We have included this point in the main manuscript.

Furthermore, the combined p-values leverage signals in the compositional “immune cell” space and absolute “tissue” space for improved power. For example, in the genetic analysis, performing Empirical Brown’s method on just the absolute results (not including the CIBERSORT-relative analysis) returns 21 significant phenotypes. This is fewer significant

findings than additionally including the CIBERSORT-Relative analysis (31). (But still more than each of the separate analyses (15, 16, 16).) We include this point in the supplement.

Lastly, we use Venn diagrams (Figures 4a, 5a-b) to demonstrate how the Empirical Brown's method leverages statistical signals across analyses. Importantly, Figures 4a and 5a show how the significant combined results are often the overlap between separate analyses. Figure 5b shows the importance of Empirical Brown's method, since variants rarely reach genome-wide significance in multiple separate analyses ($P < 5e-8$). Instead, as shown by Figure 5a, most variants with Empirical Brown's method $P < 5e-8$ are identified because they are variants associated in multiple separate analyses at $P < 1e-5$. Figure 5b shows that there are several associations identified in only one separate analysis and were not identified as significant by Empirical Brown's method, likely due to inconsistent associations across the separate analyses. We include our venn diagram-based analysis within the main manuscript, and in Figures 4a, 5a, and 5b.

3. Although grouping immune cell types into major lineages is understandable, the authors omit several key tissue-associated immune cell types from their correlative analysis without explanation (B cells, NK cells, dendritic cells, mast cells). This unexpected omission needs to be addressed as the current work is incomplete without them.

We have now expanded our analysis to 14 distinct immune cell types (CD8+ T cells, CD4+ naive T cells, CD4+ memory T cells, helper T cells, regulatory T cells, gamma delta T cells, B cells, NK cells, neutrophils, macrophages, dendritic cells, mast cells, monocytes, and eosinophils) and 2 broader immune cell types (lymphocytes and myeloid cells). This has led to substantial changes to our entire results section, as the analysis has now nearly tripled in size. We have identified new top iQTLs and our functional follow-up of eQTLs and network analysis result in new enrichments. The majority of the changes involve the changing of numbers (eg. # significant associations) within the manuscript and discussion of genetic associations within the "Association of genetic variants with infiltrating immune cells" section.

4. Age-related changes in the immune system have been previously described, particularly in the bone marrow and blood. For example, hematopoietic stem cells become myeloid-based with age (PMID 22123971). As a control, did the authors find evidence for myeloid-lymphoid skewing in aged blood by their deconvolution pipeline (e.g., as seen here in Fig 2 in PMID 26808160)?

We thank the reviewer for this important comment. As a result, we performed an in-depth analysis of whole blood to replicate previous findings with age. We discuss the results of this new analysis in a Supplementary Note. To recap, our results do indeed show evidence of myeloid-biased skewing in whole blood. We also identified an increase in NK cells and increase in CD4:CD8 T cell ratio with age, corroborating previous findings.

Specifically, the new text we have added to the main manuscript is located at the conclusion of the “Association of age and sex with immune infiltration”:

“Lastly, while we identify many new sex- and age-associated changes in tissue immunity, an analysis of aged blood samples support previous findings from other studies, including myeloid-biased differentiation¹, a rise in NK cells^{2,3}, and a decline in the ratio of CD4 to CD8 T cells⁴ (see Supplementary Note).”

Within the supplementary text, we have added the following new text under the section title “Findings in aged blood support previous research”:

“Compositional differences have been previously observed in aged blood, including myeloid-biased differentiation¹, a decline in the ratio of CD4 to CD8 T cells⁴, and a rise in NK cells^{2,3}. While our original analysis also identified an increase in NK cells ($P = 0.028$; not significant after FDR correction), we additionally calculated CD4:CD8 T cell and Myeloid:Lymphoid cell ratios in whole blood samples using CIBERSORT and xCell deconvolution estimates. We performed a rank-inverse normal transformation on these ratios to minimize outlier influence and analyzed these ratios using a similar regression model. Our model identified decreased CD4:CD8 T cell ratio and myeloid-skewing in aged blood (Age-CD4:CD8 T cells ratio association: Empirical Brown’s $P = 0.035$; Age-Myeloid:Lymphoid cell ratio association: Empirical Brown’s $P = 0.024$).”

5. Given substantial differences in mammary tissue biology between males and females, the identification of sex-specific differences in immune content in breast tissue is not surprising. A tissue type with less obvious differences in fundamental biology would be preferred.

We agree that the identification of sex-specific differences in immune content in breast tissue is not surprising, although to the best of our knowledge, it had never been reported. In addition to noting the very significant increase in lymphocytes in female breast tissue, we found significant lymphocyte differences in female thyroid tissue compared to male. We note that many thyroid diseases (autoimmune, cancer) are much more prevalent in females compared to males within the main text and include Figure 4b-c visualizing the relationship between thyroid lymphocytes and sex.

6. In the analysis of DEGs between “hot” and “cold” tissues, the authors identify a preponderance of immune-related genes, as expected. An analysis that controls for immune cell composition via a linear model would better capture expression differences that are not a surrogate for total immune cell content.

We note that our initial “hot” vs “cold” analysis including removing all genes used by our deconvolution methods to identify “hot” samples. However, as per Reviewer #1’s suggestion, we re-performed the analysis of DEGs using the xCell enrichment score as a covariate in the *limma* linear model. In our results from 10 CD8+ T cell infiltration phenotypes, we identified DEGs in different 7 phenotypes (after filtering for significance cutoffs and removal of signature genes in xCell and CIBERSORT, as described in our methods). 4 of these 7 phenotypes yielded nearly identical results in the linear model analysis: > 90% of DEGs were the same between the original results and the new xCell-covariate results. For the three other phenotypes, the top DEGs involved many immunoglobulin genes or other immune-related genes not present in the reference set from xCell and Cibersort. These results were similar for other cell types. Due to a lack of clear differences in results, we decided not to include the results in the revised version of the manuscript.

7. The infiltration phenotypes used extensively throughout this work are very unclear and their derivation is poorly described. This makes it difficult to assess the analyses and conclusions that rely on these signatures.

To improve clarity, we have made several changes:

(1) We have revised our description of what these infiltration phenotypes are and how they are derived within the main text and within the methods section.

At the conclusion of the second to last paragraph of the “Evaluating infiltration across human tissues by using deconvolution” section within the main text, we write:

“We focus on searching for these factors in a limited set of 189 filtered *infiltration phenotypes*, which represent the amount of a particular immune cell type in a specific tissue on a continuous scale.”

Within the beginning of the manuscript methods section “Merging cell subtype estimates into single scores, we write:

“From the default deconvolution scoring by CIBERSORT-Relative, CIBERSORT-Absolute, and xCell, we defined 14 specific immune cell type phenotypes (CD8+ T cells, CD4+ naive T cells, CD4+ memory T cells, helper T cells, regulatory T cells, gamma delta T cells, B cells, NK cells, neutrophils, macrophages, dendritic cells, mast cells, monocytes, and eosinophils) and 2 additional broader immune cell type phenotypes (lymphoid-based and myeloid-based cells), for a total of 16 cell types.”

And in the opening sentences of the manuscript methods section “Defining infiltration phenotypes and filtering for analysis”, we write:

“Infiltration phenotypes are a tissue-by-immune cell type pair; they are defined by the estimated amount of a particular immune cell type measured in a specific tissue on a continuous scale. As described previously, each infiltration phenotype included three separate measurements (by

CIBERSORT-Relative, CIBERSORT-Absolute, and xCell). By measuring 16 cell types in 46 GTEx tissues, which represent the 14 specific immune cell types and the 2 broader immune cell types described previously, we generated 736 original infiltration phenotypes. We then reduced to 189 more-informative phenotypes by using the filtering criteria described below:"

(2) We have also included a "terms" table for potentially unfamiliar terms used extensively in the manuscript. *Infiltration phenotypes* is included as a term in this table.

(3) We have included a Supplementary Table 18 that lists all the infiltration phenotypes derived for downstream analysis. In column 1 is tissue, and in column 2 is cell type.

8. Similarly, the approach used to delineate inflamed and non-inflamed tissues is not clearly described or graphically depicted, and the panel in figure 1d is difficult to understand. How was the infiltration z-score calculated?

Thank you for the questions regarding Figure 1D. We understand that the initial interpretation of this figure is unclear, and have revised our manuscript in response.

First, we have updated our figures to now include a graphical methods figure depicting the identification of consensus "hot" and "cold" samples after our consensus clustering procedure (Supplementary Figure 17).

Second, we have revised our methods section to more clearly describe this procedure under: *Differential expression analysis of extreme infiltration patterns*.

Lastly, the "infiltration z-score" was calculated for each phenotype by scaling the Cibersort absolute scores of each sample within the phenotype such that the set had a mean of zero and standard deviation of 1. This was done for visual purposes; specifically, such that hot/cold assignments for each phenotype could be depicted on the same scale. Otherwise, it would be difficult for readers to differentiate hot/cold clusters in phenotypes that feature very low amounts of infiltration relative to those with high means and high variances. The newest version of this plot is now in the Supplement section as Supplementary Figure 17.

Minor comments:

1. The authors conflate tissue-resident and tissue-infiltrating immune subsets. Macrophages are often tissue-resident as are resident memory T cells, mast cells, etc. While this is primarily a

semantic issue in the current work, the authors should address this issue with more nuanced language.

The reviewer is correct that we do not know for sure whether detected immune content is tissue-resident, tissue-infiltrating, or even sample (blood) contamination. To reflect this, we have altered language in the manuscript to describe infiltration as simply “immune content”. However, we continue to refer to our phenotypes as “infiltration phenotypes” for simplicity purposes.

2. An analysis of the p-values obtained by all three deconvolution techniques would be helpful to understand their concordance and how the covariance structure is being exploited by Empirical Brown’s method to produce meta-p-values.

As described in Reviewer #1’s Major Comment 2, we use Venn diagrams (Figures 4a, 5a-b) to demonstrate how the Empirical Brown’s method leverages statistical signals across analyses. Importantly, Figures 4a and 5a show how the significant combined results are often the overlap between separate analyses. Figure 5b shows the importance of Empirical Brown’s method, since variants rarely reach genome-wide significance in multiple separate analyses ($P < 5e-8$). Instead, as shown by Figure 5a, most variants with Empirical Brown’s method $P < 5e-8$ are identified because they are variants associated in multiple separate analyses at $P < 1e-5$. Figure 5b shows that there are several associations identified in only one separate analysis and were not identified as significant by Empirical Brown’s method, likely due to inconsistent associations across the separate analyses. We include our venn diagram-based analysis within the main manuscript, and in Figures 4a, 5a, and 5b.

Reviewer #2 (Remarks to the Author):

In this manuscript, Marderstein and colleagues present a retrospective study of the GTEx collection, aiming to infer determinants (genetic or otherwise) of immune infiltration. While the gene expression is provided on a tissue level, the authors utilize deconvolution tools to estimate the cellular composition that underlies the (observed) tissue average. With these estimations in hand, the authors quantify the extent of immune infiltration to various tissues and then turn to explore the relation between these estimated phenotypes and covariates of interest, and identify for instance, a significant association between the strength of neutrophil abundance in the lung and genetic variant that reside near a key transcriptional regulator. Overall, this is an important contribution, which builds on, and to some extent helps realize the potential of community efforts such as GTEx. Having said that, there are a number of caveats in the methods and in the interpretation of the results, which we describe in more detail below.

1. Stratification and analysis of “hot” vs. “cold” samples:

a. The procedure for labeling samples as “hot” or “cold” is not clearly described; please revise

the respective methods section and provide a clearer (yet short) explanation in the main text. It is my understanding is that for a given deconvolution algorithm the procedure labels *all* samples as either hot or cold. It therefore runs the risk of making arbitrary decisions when the signal is not conclusive either way (notably, this effect may be mitigated by the need to have consistent assignment across the three algorithms). How stable is this procedure? (e.g., to sub-sampling). How will it change the results (e.g., comment 1b below or on the lack of consistency in inflammation patterns across tissues) if it focuses on the “extreme phenotype” samples and excludes all others?

We apologize for any ambiguity in the description of our method. We have updated our methods section to more clearly describe the procedure for assigning samples as “hot” and “cold” within each tissue infiltration phenotype. The improved description can be found in our revised methods section in the *Differential expression analysis of extreme infiltration patterns* under **Methods**.

Additionally, we have included a graphical depiction of the procedure in Figure 3a.

As implemented, the procedure does not label *all* samples within a tissue as either hot or cold for a given deconvolution algorithm and cell type (eg. xCell in Lung - Macrophages). However, all samples will be labeled as “hot” or “cold” if the consensus clustering algorithm determines only 2 stable sub-clusters. In our expanded analysis of 189 infiltration phenotypes, this procedure was implemented 567 times (189 phenotypes x 3 deconvolution methods). Of those 567 runs, 271 yielded 2 stable sub-clusters; 282 yielded 3 subclusters, and 14 yielded 4 subclusters. After calculating mean cell type scores within each tissue, only samples within the highest and lowest clusters are labeled “hot” and “cold” respectively. In the instances with 3 or 4 stable sub-clusters, the samples within the intermediate clusters are labelled “intermediate” and were not used in downstream analysis. In the instances with 2 stable sub-clusters, not all individuals were used in differential expression analysis due to our requirement of consistency across all three deconvolution outputs. For example, samples that were labelled “Hot” by CIBERSORT-Absolute but “Cold” by xCell would not be included in the consensus “hot” or consensus “cold” sets that were used in differential expression analysis. As a result, an extreme consensus “hot” set may include only a fraction of total samples which were consistent across all three deconvolution outputs, despite an original xCell “hot” set that may have included 50% of samples (with 2 stable sub-clusters).

Nonetheless, Reviewer #2’s statement that the procedure “runs the risk of making arbitrary decisions when the signal is not conclusive” is accurate. However, as the reviewer also notes, our requirement of consistent agreement across the 3 deconvolution algorithms’ hot/cold assignments mitigates this. As per Reviewer #2’s suggestion, we demonstrated that the hot/cold assignments are robust to subsampling, by proceeding as follows. We randomly selected 6 infiltration phenotypes (Adrenal Gland – Macrophages Colon – Sigmoid – Macrophages, Colon Transverse – CD4 memory, Esophagus – Mucosa – Mast cells, Lung – Myeloid cells, and Small Intestine – B cells), and we downsampled to 50% of the total number of samples. After performing the consensus clustering procedure across the 3 deconvolution algorithms on each of the downsampled phenotypes, we found that every sample that was labeled “hot” in the subsampling was also labelled “hot” in the original assignments (with the full sample sizes). This was the same for “cold” clusters. Therefore, subsampling did not introduce any new samples into

the “hot” or “cold” clusters in our simulations. This effectively demonstrates the stability of the procedure in producing consistent assignments as hot and cold.

Furthermore, we show that similar results can also be derived by using alternative approaches for assigning samples to “immune-rich” and “immune-depleted” clusters. For each phenotype, we computed quintiles in each deconvolution output, and compared “consensus” top and bottom quintiles (sample consistently in top or bottom quintile across 3 deconvolutions for a given phenotype). The results of this differential expression analysis do not meaningfully change compared to the clustering approach. For example, in our macrophage phenotypes, the current revised analysis using consensus k-means clustering yields as top DEGs *CIQB* (18/21 tissues), *VSIG4* (17/21), *MARCO* (17/21), and *C1QC* and *CD163* (16/21). The consensus quintiles approach yields *CD163* (23/23), *C1QC*, *C1QB*, and *VSIG4* (the latter 3 all 22/23). This effect is recapitulated in the case of our CD8 T-cell phenotype as well. Our k-means approach yielded as top DEGs *CD8A*, *CD8B*, and *CD3D*, *CCL5*, and *KLRK1* (all 10/10). Using the consensus quintile approach yielded *CD8A*, *CD8B* (14/15), *CCL5* (13/15), *CD3D* (12/15), *KLRK1* (11/15).

Thirdly, we comment that our consensus k-means approach is more robust to noise intrinsic to deconvolution relative to choosing quintiles. As described by the reviewer in comment 1a, we explored a simpler approach using “consensus quintiles” to identify “extreme” samples of each phenotype. In this approach, 162/189 phenotypes featured at least 6 samples consistently identified in the top/bottom quintiles of each deconvolution. These samples were then considered for DEG analysis. However, only 136/162 phenotypes (84%) passed our DEG cutoffs. This is much lower percentage compared to our analysis using the consensus k-means approach, where 123/130 phenotypes (94.6%) passed our DEG cutoffs (dropout due to DEG failing to yield “significant outcome”). This implies that our k-means procedure is more capable of finding stable groupings that feature substantial transcriptomic differences at the outset relative to alternative methods. In so doing, the method also effectively filters noise for downstream analysis. For example, the choice of arbitrary, pre-determined cutoffs such as quintiles ignores patterns in the data distribution that can be observed by an automated clustering algorithm. If the distribution of macrophages in lung tissue were a mixture of Gaussians (with separate Gaussian parameter values for cold, intermediate, and hot), then a consensus clustering approach would discover the groupings best.

Overall, our method is complementary and perhaps an improved approach to another procedure for identifying “extreme” samples. It generally recovered the same transcriptomic differences as a quintile-based approach, without substantial loss of information after clustering relative to alternative approaches. Increased sample size would only improve the clustering algorithm because the differential expression analysis would not be impacted as much by the original sample drop out.

We have included this information within the supplement.

b. It is a bit unexpected that for ~40% (30 out of 73) of the phenotypes the differential expression analysis does not yield significant outcome, at least out of the very same sets of genes used to stratify the samples. Please provide more detail for why this test (which can be thought of as a sanity check) fails in some cases, as it might mean that the deconvolution and the resulting

stratification might be overwhelmed by noise. How correlated are the genes in each of those 30 phenotypes? Are these phenotypes discussed later on in the paper? (e.g., for association with age) and if so, would that lack of DE genes question their validity?

In the original analysis of 73 phenotypes, 23/30 of the excluded phenotypes were excluded because of insufficient number of samples to proceed with differential expression analysis. Namely, in those 23 infiltration phenotypes, there were an insufficient number of identified “hot” samples (<6 samples), which we used as our cutoff to proceed with differential expression. They were not considered for DEG, as opposed to the DEG not producing significant outcome. We apologize for the lack of clarity in the manuscript, and have revised the draft accordingly.

The revised manuscript now discusses more fully the nature of the drop out and noise in the hot/cold stratification in our expanded analysis:

Firstly, we fully agree with Reviewer #2’s concern regarding noise introduced through deconvolution (note: we provide justification for the use of deconvolution in response to comment 1d). The potential for bias and inaccuracy introduced by deconvolution methods has been characterized previously (eg. Vallania et al. 2018). Nonetheless, we effectively control for such noise through our 3 algorithms (which use two separate reference gene sets), require agreement across each, and then enforce strict cutoffs in order for results to be considered possible signal.

Secondly, we observe the same overall pattern of phenotype drop out in our expanded analysis as in the initial analysis. Our consensus hot/cold assignment algorithm yielded 130/189 phenotypes with enough samples to proceed with DEG. Of these 130 that underwent DEG, 123 pass all cutoffs (at least 5 transcripts with $\log_{2}FC \geq 2$ and $FDR < 0.01$). As with the initial analysis of 73 phenotypes, the majority of drop outs again occurs during the consensus hot/cold assignment procedure. We conclude that drop out is largely driven by our stringent requirement that an infiltration phenotype have at least 6 samples assigned to hot/cold clusters across the 3 deconvolution algorithms. We speculate that using a larger data set (eg. GTEx version 8) would result in a much smaller fraction of phenotypes that are excluded from downstream analysis due to our sample number requirements. (Of note, 21 infiltration phenotypes featured 4 or 5 consensus hot samples and were therefore excluded from DEG).

Thirdly, our consensus k-means approach is more robust to noise intrinsic to deconvolution relative to alternative methods. As described in comment 1a, we explored a simpler approach using “consensus quintiles” to identify “extreme” samples of each phenotype. In this approach, 162/189 phenotypes featured at least 6 samples consistently identified in the top/bottom quintiles of each deconvolution. These samples were then considered for DEG. However, only 136/162 phenotypes passed our DEG cutoffs. Compared to hot/cold assignments identified with consensus k-means, the quintiles approach results in far more phenotypes that dropout due to DEG failing to yield “significant outcome.” This implies that our k-means procedure is more capable of finding stable groupings that feature substantial transcriptomic differences at the outset relative to alternative methods. And in doing so, the method also filters noise from overwhelming downstream analysis.

Fourth, we identified the 66/189 phenotypes where we did not discover any differentially expressed genes and looked to see if we identified any age or sex associations. Out of these 66 phenotypes, 20 were associated with age or sex. Out of the 20, DEGs were not recovered in 18/20 simply because of insufficient sample size to proceed with differential expression. The last 2/20 failed because of strict filtering thresholds, where both phenotypes did not have at least 5 transcripts at $\log_{2}FC \geq 2.0$ and adjusted $P < 0.01$. Both have 4 DEGs with $\log_{2}FC \geq 2.0$, with 43 and 70 that are ≥ 1.5 . The 18 other phenotypes are a by-product of our assignment procedure for “immune-rich” and immune-depleted” clusters. The age and sex analysis was done separately and on continuous phenotypes (rather than “hot”/”cold”), such that the lack of clustering does not undermine the age/sex results.

Lastly, we comment on the nature of sample dropout during DEG. Of the 7/130 samples that drop out due to inability to reach DEG cutoffs, we note that these 7 infiltration phenotypes express an average of 17 signature genes (reference genes in xCell and Cibersort) at $1.0 \leq \log_{2}FC \leq 2.0$ and $FDR < 0.01$. We also note that 3/7 of these infiltration phenotypes feature the lowest variance in xCell/Cibersort Absolute scores (bottom 5 of 189) and thus their drop out due to lacking transcripts with $\log_{2}FC \geq 2$ is not unexpected. Overall, this implies that our dropout during DEG is not driven by noise, but is instead a byproduct of applying strict significance thresholds for our final results.

In summary, drop out is largely driven by phenotypes featuring insufficient number of samples to be included in DEG analysis, and not due to DEG being overwhelmed by noise intrinsic to deconvolution. For these reasons, we did not comment on them further in the manuscript. However, we have included this information within the supplementary, in the section titled “Phenotype dropout during differential expression analysis of clustered immune-rich and immune-depleted samples”.

c. “Therefore, we reflected that infiltration patterns are likely tissue-specific, rather than widespread. “ This statement may seem to be somewhat contradictory to the results in Figure 1B. Please elaborate.

In previous Figure 1B (currently Figure 2a), we focus on analyzing the between-tissue variability of immune content. We cannot assess tissue-specificity because in order to be tissue-specific, a certain tissue immune score must be compared to other tissues within the same individual. In Figure 2a, information about within-individual variability is lost by clumping into between-individual median scores. Old Figure 1b/current figure 2a is part of the “Evaluating infiltration across human tissues by using deconvolution” section. Our hypothesis regarding the tissue-specificity and sharing of the observed infiltration patterns is in a separate section of the manuscript (“Identification and characterization of extreme infiltrating immune cell patterns”). We expand on this in reviewer response comment 1E, and our hypothesis was generated based on results distinct and unrelated to old Fig 1b/curr Fig 2a, which does not inform us about tissue-specificity.

d. It is not clear from the outset why deconvolution is needed here. Would the DE genes and pathways that came up in this analysis (or other genes and pathways that make sense) be detected in a simpler correlation analysis? (i.e., correlate each gene in the transcriptome with the aggregate expression level of the genes in the same sets used for deconvolution)

As per Reviewer #2's suggestion, for a particular cell type category, we identified the signature genes used for deconvolution of that cell type. Then, for each sample, we aggregated the expression levels of the gene set by calculating the median value. This aggregated median expression level was used as a surrogate for the amount of the immune cell type in the sample. We next correlated each gene in the transcriptome against the aggregate measure. We compared the significantly correlated genes from this analysis from those that were discovered by performing a differential expression analysis of clustered "hot" and "cold" samples of the respective deconvolution scores.

By performing this analysis for macrophages within 10 different tissues, we found that the majority of highly significant, highly correlated genes corresponded to DEGs discovered in our differentially expression analysis (genes with $0.8 < \text{correlation} < 0.95$ and $10^{-80} < \text{pval} < 10^{-35}$ often corresponded to DEGs with $\log_{2}FC \geq 2$ and $FDR < 10^{-20}$). Furthermore, this approach yields our most common DEGs discussed in the manuscript as highly significant, highly correlated genes as well (eg. MARCO, CD163, VSIG4, C1QB featured $0.6 < \text{cor} < 0.95$ in many tissues, with $\text{pval} < 10^{-5}$).

Overall, we very much agree with Reviewer #2 that defining simpler metrics, such as aggregate signature gene expression levels, is a more straightforward, alternative approach to quantifying immune content in bulk samples. Previous literature has used similar methods for estimating immune content in bulk tumor sequencing⁵. However, deconvolution is preferable to the use of gene clusters/metagenes by weighting different genes separately and combining them non-linearly. As such, we chose to use both Cibersort and xCell to most accurately estimate the amounts of immune cells. The two methods are reliant on vastly different core algorithms and references for immune content estimation. Therefore, each algorithm has its own implicit biases but its own distinct strengths. In particular, Cibersort performs well with respect to measurement error and amount of unknown sample content (ie. parts of the mixture, such as tumor content, not accounted for in signature matrix)⁶. Additionally, Cibersort has shown accurate estimation of closely related cell types by accounting for multicollinearity⁷ and returns p-values that test whether any of the reference cell types were present in the input sample. On the other hand, xCell is particularly robust to any batch effects by using a ssGSEA framework^{7,8}. This is especially important in our dataset given the multi-institutional scale of the GTEx project and potential risk of technical artifacts. In addition, xCell has the highest true negative rate relative to other algorithms in simulations⁷ (eg. % of time that the deconvolution algorithm will create a score of 0 or a null score for a cell type not actually present in the sample). Therefore, leveraging both xCell and CIBERSORT was optimal to identify statistical signals.

In summary, we elected to use more sophisticated methods because of their desirable modelling properties that have shown superior performance in previous studies. We show that both the deconvolution approaches and the aggregate expression level converge on similar results. Additionally, using deconvolution can lead to more accurate and more precise results, especially in conjunction with our requirement of agreement in the results across each method. As well, we note that this aggregate expression analysis was not included in the manuscript.

e. Since the mode of number of “hot” tissues per individual x cell type was one, it is concluded that “we reflected that infiltration patterns are likely tissue-specific, rather than widespread.” This is quite a strong statement that would benefit from more context and references.

We thank the reviewer for this comment. In response, we have expanded on this analysis concerning phenotypic evidence of tissue-specificity in our revised manuscript to demonstrate greater evidence of tissue-specificity. We have included the following text to the last paragraph of “Identification and characterization of extreme infiltrating immune cell patterns”.

“

Finally, we used our immune-hot clusters (eg. macrophage-hot) to examine whether individuals with inflammation in one tissue type may also exhibit similar inflammation in their other tissue types. We analyzed individuals with at least 8 tissue samples and discovered that individuals were labeled “hot” in an average of 11-15% tissue samples per cell type, with a mode of 1 “hot” tissue per individual for a single cell type (Supplementary Figure 9). In addition, across individuals, there were no common “hot” inflammation patterns representing multiple tissues (Figure 2c; Supplementary Figure 10). Therefore, we reflected that infiltration patterns may be phenotypically tissue-specific, rather than widespread (eg. “hot”-sharing between tissues).

”

f. While the mode in the latter test is at 1 there is still a non-negligible amount of cases with more than one hot tissue. To support the validity of the results in this part of the paper (i.e., point 1e above as well as the DE analysis [point 1b]) it would be helpful to repeat the deconvolutions with random sets of signature genes per cell type (ideally, matched by mean or median expression) and compare to the resulting numbers of DE genes, numbers of hot tissues per individual, and co-clustering of similar tissues (as in Figure 1b) .

When performing CIBERSORT deconvolution, we relied on an author-provided reference matrix (LM22). This matrix consists of specific sets of signature genes that differentiate the immune cell types. As a result, we could not swap the signature genes with non-signature genes within the CIBERSORT LM22 reference profile.

Instead, we shuffled the genes within the GTEx gene expression profiles. For each signature gene (in the reference profile), we matched to another gene (via pan-tissue GTEx median gene expression values). We then replaced the gene expression values of the signature gene within the GTEx samples with the matched gene's expression values. We then performed deconvolution using the original CIBERSORT reference profile (LM22).

Overall, the “shuffled” deconvolutions results had low variability and poor detection of immune cell types. CIBERSORT computes a p-value which tests the null hypothesis that no cell types from the reference profile (22 different immune cell types) are in the sample. In the actual (original) GTEx expression profiles, 2866 of 11141 samples had CIBERSORT p-values < 0.05. In contrast, only 4 of the 11141 shuffled GTEx expression profiles had CIBERSORT p-values < 0.05. This first observation describes how CIBERSORT did not identify conclusive immune cell concentrations in nearly all shuffled GTEx expression profiles, in stark contrast to the original GTEx data. Second, heatmaps of median CIBERSORT-Absolute scores across tissues (similar to Figure 2a) in shuffled GTEx expression profiles show low variability and low values across tissue types (Review Figure 3). (The high amounts of some cell types in GTEx testis tissue can be explained by vastly different expression of some genes in GTEx testis tissue compared to other tissues, which might now be used as the “signature genes” for deconvolution.) Lastly, none of the inferred immune cell types will pass our infiltration phenotype filtering procedure described in the procedure. Therefore, the shuffled GTEx profiles result in 0 infiltration phenotypes. These observations suggest that the actual gene expression profiles result in infiltration results that are distinct from the shuffled expression profiles, and thus the shuffled expression profiles should be discarded from further analysis.

Response Figure 3: Median CIBERSORT-Absolute scores for the 14 immune cell types were calculated within each tissue. Median values were visualized in a heatmap (sorted by tissues alphabetically). Nearly all scores were close to 0 with little variability between tissues and cell types.

2. Association studies

a. Can the two significant associations be discovered with a more standard analysis, without the need for deconvolution? (namely, associate each variant with the aggregate expression level of the genes in the same sets used for deconvolution).

To address this question, we analyzed the association between rs6482199 and the Helper T cell phenotype in thyroid tissue as a representative example (and a positive result). We extracted the signature genes for the helper T cells used by CIBERSORT, calculated the median expression of these genes in the thyroid samples, and tested for the association between rs6482199 and the aggregate value. Here, we found that rs6482199 was associated with the aggregated median expression level of the helper T cell phenotype at a less significant p-value ($P = 0.0058$). We repeated this type of analysis using our second most significant association, rs56234965 with lymphocytes in sigmoid colon tissue. We identified all signature genes from all cell types in the lymphocyte phenotype, and found that the variant also correlated with the new aggregate phenotype, although less significantly so ($P = 0.00059$). These results suggest that greater resolution is obtained through deconvolution compared to a more ad hoc or simpler procedure for cell type inference, although similar results can be derived using either method.

b. An intriguing question regarding *CUX1* is whether its potential effect may be intrinsic to neutrophils or whether it relates to other cellular subsets that are in turn associated with neutrophil abundance. While this is difficult to answer, an additional discussion would be interesting. For instance, how is this gene expressed across the primary cell types in the lung? How does the chromatin look like (in terms of histone modifications, DNA methylation, accessibility in the respective loci) in the respective locus in those cells? While the ideal data needed to address this is unavailable, databases of gene expression and chromatin features per-cell types (such as Immgen) can be interesting to look at. This can further strengthen the support from refs 31 and 32. The same point holds for the second finding (rs116827016)

With the data we used for our analysis, it would be extremely interesting to infer the cell-type specific effects of genetic variation. The genetic effects could be intrinsic to the discovered cell type, more broad to other related immune cell types, more related to altering specific cell types in the primary tissue type, or have extremely broad changes to many tissue and immune cell types that alter the interaction between tissue and immune cells. However, the necessary granular cell-type expression data is available to answer this type of question is not yet available to the best of our knowledge.

First, when we analyze the gene expression for many of these genes in GTEx, Immgen, or other databases, they are rarely cell-type or even tissue-specific. But, in an analysis of *DNAJC1*, a gene identified through our top iQTL in the expanded analysis, we had a few interesting observations. First, we used GTEx to examine the *DNAJC1* gene expression across tissues (this gene was identified through an iQTL associated with the helper T cell phenotype in thyroid tissue) (Response Figure 3). We found that thyroid expression of the *DNAJC1* gene was high compared to most cell types, although bladder and lung tissues showed similar levels of expression. In contrast, whole blood expression was among the lowest. A search for *DNAJC1* in the Immgen browser did not reveal any concluding differences regarding gene expression levels between helper T cells and other cell types. Next, we searched in the protein atlas for gene expression differences and found high expression in thyroid tissue (Response Figure 4). But again, this was lower than a few other tissues, such as liver, pancreas, and prostate. *DNAJC1* thyroid expression levels was also similar to the highest blood cell types' *DNAJC1* gene expression levels. These cell types were T cells and dendritic cells. Dendritic cells are essential for antigen presentation to

helper T cells, which can then activate the adaptive immune response and stimulate B cell proliferation. Thus, this was interesting that T cells and dendritic cells had the highest expression out of the immune cells in Immgen, but the normalized expression levels were only marginally higher than the other immune cell types. Lastly, we used the protein atlas to look at *DNAJC1* protein expression across tissues (Response Figure 5). We found that expression level was rated as “high” in thyroid.

Searching rs6482199 in HaploReg found that the SNP was in strong LD with several enhancer histone marks across many cell types, including those in helper T cells, and DNase sites across many more cell types. However, there was no cell type specificity of the epigenetic marks and it is not clear how rs6482199 differences may change the epigenetic features across this region.

Overall, this type of analysis was interesting, however the results are inconclusive. The cell type expression patterns on the protein and gene level were broad with low tissue specificity. Human-derived epigenetic databases often lack the different cell types such as those in thyroid. In order to dissect the nuances of the genotypic effects across different cell types, single cell data will be necessary at scale: different individuals with different genotypes at the rs6482199 variant, along with single cell transcriptomes and single cell epigenetic (eg. ATAC-seq) information. That way, we can understand how the variant may change the molecular features within different cell types and alter the cellular interactions. Therefore, assessing cell-type specificity of SNP effects will require an expansion in single-cell -omics data that allows inference of the different cell types in the sample and their respective regulation and expression patterns.

[Redacted]

Response Figure 3: TPM gene expression for *DNAJC1* across tissues in GTEx.

[Redacted]

Response Figure 4: Normalized gene expression for *DNAJC1* across tissues in the protein atlas.

[Redacted]

Response Figure 5: Protein expression for *DNAJC1* across tissues in the protein atlas, classified as “not detected”, “low”, “medium”, or “high”.

c. It is not clear whether or not the reported cases of both i- and e- QTL were discovered in the same tissue (e.g., was the eQTL for rs11883564 computed with only sun exposed skin tissues?). If this is not the case, this analysis should be somewhat at odds with the observation that “infiltration patterns are likely tissue-specific, rather than widespread.”

We have clarified that the iQTLs and eQTLs were discovered in the same tissue within the revised manuscript. The two main ways we clarified this were to write:

(1) “Thus, we were next interested in identifying whether there were expression quantitative trait loci (eQTLs) from the GTEx consortium analysis that were also iQTLs within the same tissue (ieQTLs).”

(2) And also in the *Terms* table, where we define ieQTL as “A genetic variant that is both an eQTL and iQTL in the same tissue”.

d. To provide further support for the validity of these associations, it would helpful to repeat the analysis using random sets of signature genes per cell type, as in comment 1f above.

Please see comment 1f.

Minor comments

1. Glastonbury et al: this paper has been published. Please update the reference.

Thank you for making us aware. The reference has been updated.

2. “Therefore, these results indicate that each method provides interesting information to be exploited in downstream analysis.” Not clear how this conclusion is derived.

We have revised and expanded on this sentence at the conclusion of the “*Robust estimation of immune cell types in bulk RNA-seq profiles*” section.

“Thus, relative scores better captured compositional differences in immune content while absolute scores better captured the true cell type amount to the overall sample. Furthermore, xCell and CIBERSORT are imprecise in measuring immune content, and it could be favorable to consider where the deconvolution estimates are consistent between methods. Together, these observations indicate that the optimal analysis strategy will jointly consider CIBERSORT-Relative, CIBERSORT-Absolute, and xCell scores, equally but independently, to best capture the full range of complex patterns in deconvoluted immune content.”

3. The meaning of “CIBERSORT-Absolute” vs. “CIBERSORT-Relative” as described in the beginning of the main text is unclear and requires further reading and deciphering. Please state explicitly what these actually mean.

We have revised the text in our manuscript under the section “Robust estimation of immune cell types in bulk RNA-seq profiles” to read.

“CIBERSORT employs a linear support vector regression model to estimate cell type “relative” proportions of 22 immune cell types. Additionally, CIBERSORT calculates a scaling factor to measure the amount of total immune content in the sample, computing “absolute” scores (which are the product of the scaling factor and cellular proportions). We refer to the relative proportions from CIBERSORT as “CIBERSORT-Relative” and the product of the relative proportions with the scaling factor as “CIBERSORT-Absolute”.”

4. Figure 3 – data is colored by whether eQTLs are over- represented (red) or under-represented (blue) in the iQTLs. The terminology is a bit misleading. Over/ under- represented implies that there is statistical significance, which means that it can be (and probably often) the case that neither holds.

Since our analysis is focused entirely on over-representation of eQTLs in the iQTL results, we have modified our primary test of iQTL/eQTL overlap to be a one-sided statistical test (Figure 5i). This should also be more clear to the reader. In the supplementary, we show the results of the original two-sided test (Supplementary Figure 15). It is correct that only phenotypes above the red line indicate phenotypes with a significant over- or under-representation of ieQTL overlap, so we have modified the coloring as such. Lastly, with the expansion to 189 phenotypes, we have removed phenotype names from the plot. We did not think it was crucial to the main message of the figure, plus it did not look clear with the expansion from 73 to 189 infiltration phenotypes.

1. Pang, W.W. *et al.* Human bone marrow hematopoietic stem cells are increased in frequency and myeloid-biased with age. *Proceedings of the National Academy of Sciences* **108**, 20012-20017 (2011).
2. Valiathan, R., Ashman, M. & Asthana, D. Effects of ageing on the immune system: infants to elderly. *Scandinavian journal of immunology* **83**, 255-266 (2016).
3. Facchini, A. *et al.* Increased number of circulating Leu 11+(CD 16) large granular lymphocytes and decreased NK activity during human ageing. *Clinical and experimental immunology* **68**, 340 (1987).
4. Amadori, A. *et al.* Genetic control of the CD4/CD8 T-cell ratio in humans. *Nature medicine* **1**, 1279 (1995).
5. Bindea, G. *et al.* Spatiotemporal dynamics of intratumoral immune cells reveal the immune landscape in human cancer. *Immunity* **39**, 782-795 (2013).
6. Monaco, G. *et al.* RNA-Seq signatures normalized by mRNA abundance allow absolute deconvolution of human immune cell types. *Cell reports* **26**, 1627-1640. e7 (2019).
7. Sturm, G. *et al.* Comprehensive evaluation of transcriptome-based cell-type quantification methods for immuno-oncology. *Bioinformatics* **35**, i436-i445 (2019).
8. Aran, D., Hu, Z. & Butte, A.J. xCell: digitally portraying the tissue cellular heterogeneity landscape. *Genome biology* **18**, 220 (2017).

Reviewers' comments:

Reviewer #1 (Remarks to the Author):

Overall, the authors have done a satisfactory job addressing the majority of concerns from the prior round of review. As a result, the manuscript has improved. Nevertheless, there are still several issues that the authors should address.

Major comments:

1. The infiltration phenotypes remain very poorly described in the main text and are challenging to comprehend without a concrete definition. At a minimum, please expand the description in the main text to mirror the description in the first paragraph of "Defining infiltration phenotypes and filtering for analysis" in Methods. This is such an important part of the paper that the authors' decision to largely relegate it to Methods is perplexing.
2. The analysis of DEGs in the "extreme infiltrating immune cell patterns" section (entire 2nd paragraph) is technically flawed and overstated. Clearly, immune-hot tumors will be enriched in markers and pathways expressed by the immune cell types considered by the authors. The signatures used by cibersort and xcell are not a comprehensive compendium of leukocyte marker genes – thus, finding additional markers not present in those signatures (e.g., C1QC, FCGR3A) is expected. If the authors feel this analysis is critical, they need to more convincingly show that (i) hot samples exert context-dependent changes in leukocyte expression profiles and/or (ii) hot samples express additional genes that are not expressed by the respective immune cell types but are linked to their presence.
3. The authors have performed a new analysis showing that their deconvolution pipeline can identify age-related changes in circulating leukocytes (Supplementary Note). Please provide an accompanying figure.
4. Although the manuscript is interesting, we are disappointed with the authors' presentation. It reads as if different authors wrote different sections without making an effort to tie the material into a cohesive whole, particularly in relation to style and narrative. In addition, many of the figures are poorly constructed (e.g., Fig. 1b, Fig. 3b) and different fonts are used throughout.

Minor comments:

1. The helper T cell definition employed by the authors is confusing. Technically, Th cells are collectively defined as CD4 T cells, and this might be a clearer label to use.
2. The authors state that "xCell and CIBERSORT are imprecise in measuring immune content." Based on what reference?
3. Figure 4f,g: the direction of the effect ('+', '-') is challenging to interpret without a reference. I assume '+' refers to female in panel f but this is not explicitly stated. Does '+' imply that the effect is associated with higher age in g? Given the ordering of panels, wouldn't it make more sense to title the figure: "Significant associations with sex and age?"

Reviewer #2 (Remarks to the Author):

The authors have significantly improved the presentation of the methods and results and expanded the analysis in response to the previous reviews. There are a few questions that remain, mostly following the new exposition of methods and intermediary results:

1. Hot vs. cold analysis:

- 1a. The procedure for assignment of samples to hot and cold is now defined clearly. The clearer

definition raises questions though – Why do we “clone” the 1D vector to three identical columns? Also - I can see that the authors explored the idea of using quintiles instead, but I as far as I could see it is not mentioned in the main text. Please make sure to report this and to better rationalize and justify this nonstandard design choice.

1b. “we reflected that infiltration patterns are likely tissue-specific, rather than widespread. ” – this is quite a strong statement. Can it be the case that this extreme conclusion be somewhat mitigated when we consider “nearly significant” cases? For instance – while the mode of #hot tissues for an individual is 1, can it be the case that other tissues reach “near significance” levels in those cases (i.e., individuals with a clear evidence for infiltration)?

2. In the new suppl figure 12 (QTL analysis), the deviation from the null distribution towards higher significance seems to be only achieved by CyberSort-Abs. It is therefore somewhat confusing to see that combining all three methods yielded more significant calls than either one in isolation. Please explain this point.

3. Related to that -- can it be that the procedure for combination of P- values in the iQTL analysis somehow creates an inflation, and may thus require a more stringent null? (this can be tested by shuffling the data).

4. A p value cutoff of 5.0×10^{-8} is used to define “genome wide significance”. The meaning of this cutoff is not very clear. How does it translate to false discovery rate?

5. The top hits in the iQTL seem to have changed between this and the original submission. I am assuming that this is because the authors expanded the set of infiltration phenotypes. Can the authors please confirm that the previous results (e.g., the top three hits) have been reproduced in their reanalysis and that they are not featured in the manuscript since there are other, new and more significant, hits?

6. The “shuffled deconvolution” analysis is important to support the results. Please include it in the manuscript (methods/ supplementary with a mentioning in the main text).

7. Following the previous review, the authors have demonstrated that a simple approach (with no deconvolution) works similarly to the deconvolution- scheme in the case of hot vs. cold analysis, as well as the iQTL analysis (albeit with less power). While the authors provide some rationalization for not pursuing this simpler approach in the response letter, the explanation is somewhat lacking since it is not accompanied by any empirical evidence from this study.

While such empirical evidence would have been ideal, I think that the current analysis is sufficient.

However, in order to be completely transparent with the readers -- please include this simple analysis (hot vs. cold and iQTL) as a supplementary section and acknowledge the fact that it can recover similar trends.

Minor comments:

8. The differences between the three methods are now clearly explained. It would be nice however (but not critical), to make a better case as for why taking the consensus is a good approach. Specifically: “these observations indicate that the optimal analysis strategy will jointly consider CIBERSORT- Relative, CIBERSORT-Absolute, and xCell scores” can the authors demonstrate this point based on their pseudo- simulated analysis? (described under section “Robust estimation of immune cell types in bulk RNA-seq profiles.”)

9. Enrichment analysis of DEG genes in hot vs. cold: the authors note that while some genes are indeed members of the signature used to stratify the samples, some are new. However, this point is ignored in the enrichment analysis (IPA). Ideally the authors should repeat this analysis, while excluding the signature genes. This should be reflected in both the foreground and background gene sets used for the enrichment tests.

10. “Interestingly, recent immunological evidence suggests a clear genetic influence over tissue

infiltration [7]" Reference seems wrong

11. On the first results section and Figure 1: please explicitly describe the relationship between the 22 cell type panel (CSORT) and the 64 panel (xCell).

12. "Many tissues featured a majority of samples with **trace** immune enrichment," Typo?

13. Supplementary Table 3: please specify how much variance is captured by each PC in each of the tissues.

14. "Furthermore, we found that the top 31 iQTLs from these 31 phenotypes were significantly enriched for being a previous GWAS" please specify the fold enrichment (i.e., observed vs. expected percent of GWAS amongst the iQTLs) and P-value.

15. Y axis label in Fig 5e: I am assuming the authors are referring to the T helper cells infiltration phenotype. However, "Tfh" usually stands for T follicular helper cells (a specific subtype of CD4+ T cells).

We thank the reviewers for reading our manuscript and providing additional feedback. They each raised thought-provoking questions about the manuscript, and we believe that our new analyses have significantly improved the manuscript. In response, we have made several revisions to the manuscript. These broadly include the addition of new simulations to support our previous methods (Empirical Brown's method's false positive rates), new statistical analyses to support previous conclusions ("hot" tissue-specificity), and major structural changes to the narrative for improving manuscript readability. Our full response to the recent reviewer comments are described below. In the Supplement and Main Text, new changes are made in red.

Reviewers' comments:

Reviewer #1 (Remarks to the Author):

Overall, the authors have done a satisfactory job addressing the majority of concerns from the prior round of review. As a result, the manuscript has improved. Nevertheless, there are still several issues that the authors should address.

Major comments:

1. The infiltration phenotypes remain very poorly described in the main text and are challenging to comprehend without a concrete definition. At a minimum, please expand the description in the main text to mirror the description in the first paragraph of "Defining infiltration phenotypes and filtering for analysis" in Methods. This is such an important part of the paper that the authors' decision to largely relegate it to Methods is perplexing.

We thank the reviewer for highlighting the importance of clearly defining "infiltration phenotype" in the manuscript. We have included an expanded description in the Results section entitled "Evaluating infiltration across human tissues by using deconvolution." The description has been added when the term "infiltration phenotype" is first introduced, and now closely mirrors the definition as presented in the Methods section.

"We focused on searching for these factors in a limited set of 189 filtered *infiltration phenotypes*, which represent the amount of a particular immune cell type in a specific tissue on a continuous scale. This set was derived from an unfiltered list of 736 infiltration phenotypes, which encompass 14 specific immune cell types (described above) and 2 broader cell types (lymphoid & myeloid) in the 46 GTEx tissues, as estimated across three separate measurements (by CIBERSORT-Relative, CIBERSORT-Absolute, and xCell). The set was filtered down to 189 phenotypes, which was performed using several criteria that considered sufficient immune content and correlated estimations between xCell and CIBERSORT (see **Methods**)."

2. The analysis of DEGs in the "extreme infiltrating immune cell patterns" section (entire 2nd paragraph) is technically flawed and overstated. Clearly, immune-hot tumors will be enriched in markers and pathways expressed by the immune cell types considered by the authors. The signatures used by cibersort and xcell are not a comprehensive compendium of leukocyte marker genes – thus, finding additional markers not present in those signatures (e.g., C1QC, FCGR3A) is expected. If the authors feel this analysis is critical, they need to more convincingly show that (i) hot samples exert context-dependent changes in leukocyte expression profiles and/or (ii) hot samples express additional genes that are not expressed by the respective immune cell types but are linked to their presence.

We agree with the reviewer that the reference gene sets in xCell and Cibersort are not comprehensive, and the presence of additional markers is to be expected in the differential expression analysis. We also agree with the reviewer that the hot/cold DGE results feature a paucity of evidence of 1) context-specific differences in leukocyte expression profiles, and 2) definitive determination of additional gene expression differences between hot/cold samples in tissue parenchyma. We were unable to unambiguously detect such tissue-specific transcriptomic drivers of immune expression or immune cell-type specific changes in gene expression.

Overall, we hoped to acknowledge these shortcomings in the previous version of the manuscript. However, based on the reviewer's comments, we have decided to overhaul major features of this section in the Results, by moving many results from the main text to the Supplement. We have retained in the main manuscript the content discussing the consensus k-means method, and the implications of our findings regarding whether individuals were identified as "hot" across multiple tissues. We have also added additional discussion of motivation for this analysis. Regarding the results of the differential expression, we have more explicitly commented that our analysis was unrevealing of major pathways or transcripts that can be considered tissue-specific determinants of baseline immune infiltration.

In addition, we have introduced in the main text an alternative analysis to the consensus k-means clustering approach, consisting of exploring “consensus” quintiles for each infiltration phenotype (this analysis was previously described in the Supplement but not mentioned in the main text). For each infiltration phenotype, we computed quintiles in each deconvolution output, and compared “consensus” top and bottom quintiles (sample consistently in top or bottom quintile across 3 deconvolutions for a given phenotype). The results of this differential expression analysis do not meaningfully change compared to our clustering approach. We comment extensively in the Supplement on why we believe the k-means procedure is more capable of finding stable groupings that feature substantial transcriptomic differences at the outset, and therefore more effectively filters noise for downstream analysis. We believe that this demonstrates the efficacy of this method, by producing nearly identical results to alternative methods and does not suffer from the introduction of arbitrary cutoffs.

3. The authors have performed a new analysis showing that their deconvolution pipeline can identify age-related changes in circulating leukocytes (Supplementary Note). Please provide an accompanying figure.

Our analysis identified a statistically significant age-related increase in the ratio of Myeloid:Lymphoid cells ($P = 0.024$). In the newest version of the manuscript, we have included a figure panel to accompany this analysis (Supplementary Figure 18). In this figure, we first show how the CIBERSORT and xCell-based estimates for Myeloid:Lymphoid Ratio are correlated ($r = 0.54$). Next, we show the relationship between Age and Myeloid:Lymphoid Ratio for xCell. This analysis has been added to the methods section and the accompanying computer code is included on our GitHub page.

Supplementary Figure 18: Myeloid:Lymphoid ratios were calculated in whole blood and adjusted for linear model covariates, minus age (see **Methods**). Left, the myeloid:lymphoid phenotype from xCell is plot against the phenotype derived from CIBERSORT. Right, the relationship between age and myeloid:lymphoid ratio in xCell. *BETA* represents the effect size β from the linear model between age (numerical; discrete, binned into 10-year categories) and myeloid:lymphoid ratio. Myeloid:lymphoid values are covariate-adjusted.

4. Although the manuscript is interesting, we are disappointed with the authors’ presentation. It reads as if different authors wrote different sections without making an effort to tie the material into a cohesive whole, particularly in relation to style and narrative. In addition, many of the figures are poorly constructed (e.g., Fig. 1b, Fig. 3b) and different fonts are used throughout.

We appreciate the reviewer’s concerns regarding the flow and cohesion of the prose within the manuscript. In the current version of the text, we made several revisions to improve presentation of our results.

- (1) We have posted the primary goals of performing an analysis in the beginning of paragraphs to improve the understanding of our motivation.
- (2) We split up the genetic analysis results section into three smaller sections.

(3) We have tried to improve the flow from one section into another, such as introducing the hot/cold analysis as our approach to more precisely explore the transcriptomic differences among samples after our prior finding of significant variation in bulk expression being driven by immune content

(4) We have also worked to improve the writing, and hope this better accomplishes a more cohesive narrative and improved clarity throughout the manuscript.

We also thank the reviewer for calling our attention to the inconsistency in figure construction. We have amended the problematic figures by altering the font to Helvetica.

Minor comments:

1. The helper T cell definition employed by the authors is confusing. Technically, Th cells are collectively defined as CD4 T cells, and this might be a clearer label to use.

We thank the reviewer for identifying a potential source of confusion. Our distinction of the T helper cell phenotype is meant to isolate and study infiltration of fully differentiated Th1/Th2 and CD4+ follicular helper cells. We recognize the value of more explicitly emphasizing this distinction within the manuscript. To clarify this, within the main text, we have written: “The most significant iQTL we identified was an association between rs6482199 and helper T cells (in particular, Th1, Th2, and T follicular helper cell content inferred by the deconvolution algorithms) in thyroid samples ($P = 7.5 \times 10^{-10}$) (Figure 5c-d).” This is located in the second paragraph of the section “Association of genetic variants with infiltrating immune cells.” We reserved the explicit label “CD4+ T cell” for our analysis of CD4:CD8 ratio, and in this setting, the “CD4+ T cell” label corresponds to all CD4-bearing cells (including helper T cells, Tregs, and memory CD4+ T cells). We have made this distinction in the section “Association of age and sex with immune infiltration”: “Lastly, while we identify many new sex- and age-associated changes in tissue immunity, an analysis of aged blood samples support previous findings from other studies, including myeloid-biased differentiation¹ (Supplementary Figure 18), a rise in NK cells^{2,3}, and a decline in the ratio of CD4 to CD8 T cells⁴ (defined as all CD4 and CD8 bearing T cells; see Methods and Supplementary Note for results).”

2. The authors state that “xCell and CIBERSORT are imprecise in measuring immune content.” Based on what reference?

This comment is a conclusion from the results in our simulations, as described in the “Robust estimation of immune cell types in bulk RNA-seq profiles” section. In the previous simulations, we showed that xCell and CIBERSORT scores both correlate strongly with the true immune cell type amounts, and correlate with each other, but there are differences (correlation coefficient r is *not* equal to 1, and see new supplementary figure 19 for a case example of CIBERSORT vs xCell differences on two samples with same true immune cell counts).

In addition, recent literature has explored the strengths and limitations of various deconvolution methods⁵⁻⁷. In review, CIBERSORT performs well with respect to measurement error and amount of unknown sample content (which describe parts of the mixture, such as tumor content, not accounted for in the signature matrix)⁵. And further, benchmarks have shown it is capable of accurate estimation of closely related cell types by accounting for multicollinearity⁶. However, other studies have demonstrated that deconvolution methods like CIBERSORT are critically dependent on the choice of reference expression matrix. CIBERSORT, in particular, has been demonstrated as introducing biological and technical biases in its deconvolution due to the fact that its default reference matrix LM22 (which was used in our study) was created using only healthy samples from a single microarray platform⁷. A major advantage of xCell is that it is particularly robust to any batch effects by using a ssGSEA framework⁶. However, like CIBERSORT, its accuracy is dependent on its reference gene matrix. Furthermore, when first testing these methods, we found that the output of xCell is dependent on all samples that are analyzed in a single run. Because xCell uses the variability among the samples for a linear transformation of the output score, a given sample can have different scores when submitted together with other samples⁶. With this in mind, in addition to our own simulation analyses, we concluded that the most accurate results from downstream analysis would entail immune content estimates from xCell and CIBERSORT jointly. We describe some of these observations from the previous literature in the supplementary note, “**Results from using an aggregate expression analysis versus deconvolution**”.

Overall, we have revised the paragraph including this sentence to be more specific:

“Thus, our analyses revealed that relative scores better captured compositional differences in immune content, while absolute scores better captured the true cell type amount to the overall sample. Furthermore, while xCell-based and CIBERSORT-based estimates correlate with true immune cell amounts and with each other, there is not perfect correlation (see Supplementary Figure 19 for a case example where xCell and CIBERSORT differ in simulated data). As a result, an effect or difference may be better captured and detected in one deconvolution method compared to another. These observations indicate that it could be favorable to leverage information across deconvolution estimates. Furthermore, they suggest that the optimal analysis strategy will jointly consider CIBERSORT-Relative, CIBERSORT-Absolute, and xCell scores, equally but independently, to best capture the full range of deconvoluted immune content (see Supplementary Note).”

3. Figure 4f,g: the direction of the effect ('+', '-') is challenging to interpret without a reference. I assume '+' refers to female in panel f but this is not explicitly stated. Does '+' imply that the effect is associated with higher age in g? Given the ordering of panels, wouldn't it make more sense to title the figure: “Significant associations with sex and age?”

The interpretation of +/- was lost when transferring from table format (original submission) to figure format (current submission). We thank the reviewer for catching this and have updated the manuscript.

Within figure caption:

“(f-g) Summary of (f) sex and (g) age association results from all 189 infiltration phenotypes. +/- indicates effect direction from CIBERSORT-Absolute analysis (increase in females or higher age)”

We have also updated the figure title to “Significant associations with sex and age” as suggested.

Reviewer #2 (Remarks to the Author):

The authors have significantly improved the presentation of the methods and results and expanded the analysis in response to the previous reviews. There are a few questions that remain, mostly following the new exposition of methods and intermediary results:

1. Hot vs. cold analysis:

1a. The procedure for assignment of samples to hot and cold is now defined clearly. The clearer definition raises questions though – Why do we “clone” the 1D vector to three identical columns? Also - I can see that the authors explored the idea of using quintiles instead, but I as far as I could see it is not mentioned in the main text. Please make sure to report this and to better rationalize and justify this nonstandard design choice.

We thank the Reviewer for identifying the sentence in the Methods regarding the consensus clustering procedure. The single 1 x N vector was replicated into a 3 x N matrix (with 3 identical rows, not columns). This was done to format the data properly for the consensus clustering algorithm within R. We apologize for any confusion and have clarified this in the manuscript. In response to the reviewer's comment, we have also more explicitly discussed the quintiles analysis and our reasoning behind using the consensus clustering approach. We have included an explicit reference to our Supplement for readers to consult the details of our comparison of the two methods, and our reasoning that the clustering approach is better powered to uncover stable groupings and more effectively filter noise from downstream analysis. We have also included the details of IPA on filtered genes, which we discuss on further in response to comment 9.

As well, after discussions between the authors and feedback from Reviewer #1 and Reviewer #2, we have decided to move the details of the differential expressed gene analysis to the supplement.

1b. “we reflected that infiltration patterns are likely tissue-specific, rather than widespread.” – this is quite a strong statement. Can it be the case that this extreme conclusion be somewhat mitigated when we consider “nearly significant” cases? For instance – while the mode of #hot tissues for an individual is 1, can it be the case that other tissues reach “near significance” levels in those cases (i.e., individuals with a clear evidence for infiltration)?

In the newest version of the manuscript, we introduce a statistical test to further assess tissue-specificity of infiltration patterns.

From the methods:

Tissue-specificity of infiltration patterns

We explored whether individuals “hot” in one tissue type were more likely to be “hot” in other tissue types. For each cell type, all individuals with at least 8 tissue samples represented within the infiltration phenotypes (for that cell type) were identified. The median and mode number of “hot” tissues within these individuals were calculated. Hierarchical clustering was performed between tissues and individuals, where binary values represent “hot” or “not hot” in a particular tissue for each individual.

To formally analyze whether “hot” patterns in one tissue are independent of “hot” patterns in other tissues, the immune-hot clusters from the infiltration phenotypes were assessed using a Fisher exact test. This was performed as follows. First, for a particular cell type, all tissues used within the 189 infiltration phenotypes were identified. Next, for each possible pair of these tissues, all individuals who contributed samples to both tissue types were identified. A two-by-two contingency table was then created for each tissue pair, where samples are classified as “hot” or “not hot” in each tissue. Finally, a Fisher exact test was used to assess the null hypothesis that the two tissues exhibited independent “hot” sharing patterns. Non-independent “hot” sharing patterns indicates that the probability of one tissue being inflamed is conditional on another tissue being inflamed. This process was repeated across tissue pairs for all cell types, and a Bonferroni correction was used to assess significance. The procedure was performed on “hot” clusters based on top 20% scores across all three deconvolution methods (quintiles), top 40% scores across all three deconvolution methods (top two quintiles), or consensus clustering (described above in greater detail).

From the main text:

Finally, we used our immune-hot clusters (*e.g.* macrophage-hot) to examine whether individuals with inflammation in one tissue type may also exhibit similar inflammation in their other tissue types. For each cell type, we analyzed the distribution of “hot” tissues across individuals with at least 8 different tissue samples. Within the consensus clusters, we discovered that individuals were labeled “hot” in an average of 9.1-12.5% tissue samples per cell type, with a mode of 1 “hot” tissue per individual for a single cell type (Supplementary Figure 9) (within quintile clusters, individuals were labeled “hot” in an average of 16.6-25.0% tissue samples). In addition, across individuals, there were no clear, common “hot” inflammation patterns representing multiple tissues (Figure 3b; Supplementary Figure 10). We developed a statistical method to formally test the hypothesis of independent “hot” inflammation patterns between any two tissues (see **Methods**). This hypothesis could not be rejected in any tissue pair using the consensus clusters ($P < 2.8 \times 10^{-5}$); however, by using the quintile-based clusters, we found evidence ($P < 2.8 \times 10^{-5}$) for an association between “hot” lung and “hot” whole blood samples for CD8+ T cell content ($P = 2.7 \times 10^{-6}$; in the consensus clustering analysis, we found a P -value = 1.9×10^{-3} ; Supplementary Figure 21). We next relaxed the “hot” cluster requirements to now include individuals within the top two quintiles (40%) across all three deconvolution methods, which comprise individuals at above average but not necessarily extreme immune cell levels. Using these new clusters, we found 9 pairs of tissues with significant associations ($P < 2.8 \times 10^{-5}$) for particular immune cells (out of 1796 tissue pairs tested) (see Supplementary Table 20). Therefore, we note that extreme infiltration patterns appear to generally be phenotypically tissue-specific, rather than widespread (*e.g.* “hot”-sharing between tissues). However, when assumptions were relaxed to reflect above average rather than extreme immune content, there appeared to be evidence for particular pairs of tissues with some level of shared immunity.

From the supplement:

Supplementary Table 20: Significant results from assessing the sharing of “hot” patterns between two tissues. Clusters are based on the top two quintiles (40%) approach, and p-values are based on Fisher’s exact p-values.

<https://drive.google.com/drive/u/2/folders/1veU3GKZqvb1mI-aH6BGPCHGuMRmyiOLr>
See *SuppTab19.txt*.

Supplementary Figure 21: Heatmaps display “hot” patterns across individuals with both lung and whole blood samples. If individuals do not have both whole blood and lung samples, individuals were removed from further

analysis. “Hot” groupings were determined by (a) consensus k-means clustering, (b) quintiles, and (c) top two quintiles. Rows represent tissues and columns represent individuals. Red indicates the individual was labeled “hot” in that tissue type, while blue represents not “hot” (intermediate or cold). Columns (individuals) were clustered by Euclidean distance.

2. In the new suppl figure 12 (QTL analysis), the deviation from the null distribution towards higher significance seems to be only achieved by CyberSort-Abs. It is therefore somewhat confusing to see that combining all three methods yielded more significant calls than either one in isolation. Please explain this point.

In our paper, we describe how each of our deconvolution measurements (xCell, CIBERSORT-Relative, CIBERSORT-Absolute) have different estimates of the immune cells within the samples, and that the ideal analysis should take into account each of the different deconvolution measurements. Our approach is to use Empirical Brown’s method to combine p-values from all three analyses in each of the deconvolution scores separately. In Figure 5a, we show that of the 31 infiltration phenotypes where we identified at least one iQTL with Empirical Brown’s $P < 5 \times 10^{-8}$, 23 had $P < 1.0 \times 10^{-5}$ across xCell, CIBERSORT-Relative, and CIBERSORT-Absolute analyses. Furthermore, 12 of these 31 phenotypes had $P > 5e-8$ in each of the separate analyses (as shown by Figure 5b). Our interpretation of this observation is that Empirical Brown’s method assigns “higher confidence” to where the results are consistent across the different deconvolution methods, reducing the bias from selecting a single method and improving power by valuing consistency. In these 12 phenotypes, the p-values under a single deconvolution analysis never reached genome-wide significance but, together, our method boosts the p-value below $5e-8$. We also show in Figure 5b, there are 9, 5, or 4 phenotypes where there was an iQTL associated at $P < 5e-8$ in a separate analysis in CIBERSORT-Relative, CIBERSORT-Absolute, or xCell respectively but not in any other phenotype, and not in the combined Empirical Brown’s p-value analysis. By choosing a “consensus” p-value, we can be more confident about the association and less worried about any estimation error induced by the choosing a particular deconvolution method. In the case of the specific iQTL in question (rs648299), the separate p-values of $5.7e-7$, $3.29e-9$, and $9.8e-4$ resulted in an Empirical Brown’s p-value = $7.5e-10$.

3. Related to that -- can it be that the procedure for combination of P- values in the iQTL analysis somehow creates an inflation, and may thus require a more stringent null? (this can be tested by shuffling the data).

To test whether Empirical Brown’s method (EBM) could inflate P -values in our study, we performed shuffled data experiments using our original finding of rs648299 and its association with the helper T cell phenotype in thyroid tissue. In the original study using real data, this SNP has an EBM P -value = 7.5×10^{-10} , derived from P -values = 5.7×10^{-7} , 3.3×10^{-9} , and 9.8×10^{-4} in CIBERSORT-Rel, CIBERSORT-Abs, and xCell separate analyses.

In our implemented shuffling procedure, we sampled the covariate-adjusted phenotypic values for the helper T cell phenotype, without replacement between individuals but identically across deconvolution methods and within individuals. To clarify, this means that each phenotypic value will be assigned to a new individual, and individual i who is assigned the CIBERSORT-Rel value from individual j will also be assigned the CIBERSORT-Abs and xCell values from individual j . In this way, the covariance matrix between CIBERSORT-Rel, CIBERSORT-Abs, and xCell score will be preserved and identical, which is used in Empirical Brown’s method. We next analyzed the association between the original rs648299 and the new, shuffled phenotype for CIBERSORT-Rel, CIBERSORT-Abs, and xCell using a simple linear model. The P -values from the three linear models were combined using EBM. This process was repeated for 10,000 simulations, and P -value inflation was assessed by analyzing false positive rate. False positive rate was calculated by identifying the percentage of simulations where $P < 0.05$.

Our results found a false positive rate of 0.0529, 0.0504, and 0.0483 for CIBERSORT-Rel, CIBERSORT-Abs, and xCell separate analyses. Using the combined EBM framework, the false positive rate was lower, 0.0388. We also note that there was a single simulation where CIBERSORT-Absolute analysis returned a P -value of 1.9×10^{-6} , which is lower than a Bonferroni-corrected P -value threshold of $P < 0.05/10000 = 5.0 \times 10^{-6}$. In this case, analysis in CIBERSORT-Relative returned $P = 7.3 \times 10^{-3}$ and analysis in xCell returned $P = 4.4 \times 10^{-2}$. The combined EBM P -value was 3.6×10^{-5} , which is greater than the $P < 5.0 \times 10^{-6}$ Bonferroni cut-off. Thus, the combined p-value framework helped decrease the significance of this P -value such that it would not be rejected under a Bonferroni correction by leveraging the lower strength of association within the other deconvolution method analyses.

Overall, in this shuffled data experiment, there did not appear to be P -value inflation.

We have added this analysis as a supplementary section in **“Using empirical data to analyze potential inflation in Empirical Brown’s P -values.”** In the main text, we mention:

“The most significant iQTL we identified was an association between rs6482199 and helper T cells (in particular, Th1, Th2, and T follicular helper cell content inferred by the deconvolution algorithms) in thyroid samples ($P = 7.5 \times 10^{-10}$) (Figure 5c-d). **We conducted simulations to examine the false positive rate between this SNP and the phenotype, but found no evidence for P -value inflation (see Supplementary Note).**”

4. A p value cutoff of 5.0×10^{-8} is used to define “genome wide significance”. The meaning of this cutoff is not very clear. How does it translate to false discovery rate?

There are many methods used for multiple hypothesis testing corrections: Benjamini-Hochberg, q-values, Bonferroni, etc. One issue in their implementation within genetic analysis has been the high correlation between individual SNPs in the human genome due to linkage disequilibrium. This results in a much fewer number of distinct regions in the genome that are being tested compared to the number of different SNPs actually being tested (since each SNP is not independent from another). As a result, it has been “unofficially” adopted by much of the GWAS literature to use a genome-wide significance threshold of $P < 5.0 \times 10^{-8}$, which is roughly derived from a Bonferroni correction over approximately 1 million distinct regions in the genome (broken up by recombination)⁸. When preparing our manuscript, we carefully discussed a significance threshold to report. At that time, after reviewing some of the most recent literature in *Nature Genetics*⁹⁻¹⁴, we decided to report the $P < 5.0 \times 10^{-8}$ threshold within our final analysis. We believe that false discovery rate, Bonferroni correction, q-values, and other approaches are also appropriate, but that a p value cutoff of 5.0×10^{-8} is suitable and in line with multiple hypothesis corrections currently used within the majority of GWAS literature.

5. The top hits in the iQTL seem to have changed between this and the original submission. I am assuming that this is because the authors expanded the set of infiltration phenotypes. Can the authors please confirm that the previous results (e.g., the top three hits) have been reproduced in their reanalysis and that they are not featured in the manuscript since there are other, new and more significant, hits?

The top iQTLs hits are different between the two analyses as the new analysis has been expanded and produced new, more significant iQTLs. In our new analysis, the previous most significant iQTL, rs77155650, has $P = 2.80 \times 10^{-6}$, and the previous 2nd most significant iQTL, rs116827016, has $P = 2.83 \times 10^{-3}$. The infiltration phenotype for the third most significant iQTL, CD4+ T cell in sun exposed skin tissue, was removed due to the split of the CD4 T cell category into CD4 T cell subtypes.

6. The “shuffled deconvolution” analysis is important to support the results. Please include it in the manuscript (methods/ supplementary with a mentioning in the main text).

We have added the following sentence to the first paragraph of the “Evaluating infiltration across human tissues by using deconvolution” section:

“We note that deconvolution of the true GTEx gene expression profiles produced very distinct estimates compared to “control” samples where the gene expression profile was shuffled (see Supplementary Note and Supplementary Figure 20).”

In the supplementary note, we have a section titled “Control: Influence of random genes on deconvolution estimates” that describes the previously discussed analysis in detail (from the previous review).

7. Following the previous review, the authors have demonstrated that a simple approach (with no deconvolution) works similarly to the deconvolution- scheme in the case of hot vs. cold analysis, as well as the iQTL analysis (albeit with less power). While the authors provide some rationalization for not pursuing this simpler approach in the response letter, the explanation is somewhat lacking since it is not accompanied by any empirical evidence from this study. While such empirical evidence would have been ideal, I think that the current analysis is sufficient. However, in order to be completely transparent with the readers -- please include this simple analysis (hot vs. cold and iQTL) as a supplementary section and acknowledge the fact that it can recover similar trends.

We have included a description of our previous analysis within the supplementary section titled “Results from using an aggregate expression analysis versus deconvolution”.

Minor comments:

8. The differences between the three methods are now clearly explained. It would be nice however (but not critical), to make a better case as for why taking the consensus is a good approach. Specifically: “these observations indicate that the optimal analysis strategy will jointly consider CIBERSORT- Relative, CIBERSORT-Absolute, and xCell scores” can the authors demonstrate this point based on their pseudo- simulated analysis? (described under section “Robust estimation of immune cell types in bulk RNA-seq profiles.”)

We have added a new supplementary section titled “**Analyzing the utilization of multiple deconvolution methods in simulations**”, where we assess power and false positive rates using pure simulations and synthetic mix data. This section has been copied below. We link to it within the final paragraph of the “**Robust estimation of immune cell types in bulk RNA-seq profiles**” section by a referral to the supplementary note.

In the main text, we describe how xCell and CIBERSORT calculate scores that correlate with the true amounts and correlate with each other, but do not perfectly correlate with each other. In Supplementary Figure 19, we show how even in simulated synthetic mixes where the true amounts are known to be equal, the immune cell estimates can differ. For example, in this particular scenario, CIBERSORT-Absolute correctly detects that CD4+ T cell content is equal between the two simulated samples. In contrast, xCell produces an estimate for Sample 2 that is roughly 60% of Sample 1. Therefore, the overall correlation results in our simulations have shown that these methods describe alternative yet reasonably accurate perspectives of immune cells in the test sample; but, there are clear cases where there exist differences. These differences can lead to *effect size heterogeneity*, where an effect may be better detected in one deconvolution method compared to another, such as xCell versus CIBERSORT (due to biases from different algorithms or reference matrices) or Relative versus Absolute (does the genetic effect alter the composition of immune cells in the sample or does the genetic effect alter the total amount of a particular immune cell in the sample?). To identify associations, we hypothesized that it makes sense to utilize this statistical heterogeneity within analyses by leveraging shared signals across deconvolutions rather than ignoring it by choosing a single deconvolution method.

To test this hypothesis using our previously simulated $N=80$ synthetic samples, we consider a SNP analysis. We simulated a SNP genotype with $MAF = 0.4$ in $N=80$ individuals, coded as 0, 1, or 2 from a binomial distribution. In the causal scenario, we let the SNP randomly explain 0 – 8% variance in each of the three deconvolution phenotypes (CIBERSORT-Abs, CIBERSORT-Rel, and xCell) by (1) randomly selecting three values from a uniform distribution between 0 and 0.08 to allow effect size heterogeneity across the deconvolution outputs, and (2) rescaling the genotypic (binomial) and environmental (original deconvolution score) components. In the non-causal scenario, the SNP contributes 0% variance to each of the deconvolution phenotypes (thus, phenotypes used are the original deconvolution scores). We then tested the association between the SNP and phenotype for each deconvolution method using a linear model, merged these P -values using the combined Empirical Brown’s method, and rejected the null hypothesis of no association when $P < 0.05$. We repeated this 10,000 times in the causal scenario and 10,000 times in the non-causal scenario. We found that the power in the causal scenarios was 0.5164 using the Empirical Brown’s method, but 0.4585, 0.4664, and 0.4662 in the xCell, CIBERSORT-Rel, and CIBERSORT-Abs separate analyses. Furthermore, we found that the false positive rate increase in the non-causal scenario was negligible: 0.0555 using the Empirical Brown’s method, and 0.0517, 0.0509, and 0.0500 in the xCell, CIBERSORT-Rel, and CIBERSORT-Abs separate analyses. Finally, our analyses found zero non-causal SNPs with $P < 5 \times 10^{-8}$, but 5 of

10,000 using the EBM *P*-values, compared to 3, 1, and 2 in the separate analyses from xCell, CIBERSORT-Rel, and CIBERSORT-Abs. Therefore, the combined approach using Empirical Brown's method revealed superior power while maintaining low false positive rates as compared to the separate analyses.

Supplementary Figure 19: Two synthetic samples with 10% CD4 T cell content. Sample 1 is in blue, sample 2 is in orange. CD4+ T cell content was estimated in both samples using xCell and CIBERSORT-Absolute. Along the y-axis, the relative difference in scores to sample 1 for each deconvolution method are shown: (Estimated CD4+ T cell score in Sample 1 / Estimated CD4+ T cell score in Sample K; therefore, sample 1 (blue) has a relative difference of 1).

9. Enrichment analysis of DEG genes in hot vs. cold: the authors note that while some genes are indeed members of the signature used to stratify the samples, some are new. However, this point is ignored in the enrichment analysis (IPA). Ideally the authors should repeat this analysis, while excluding the signature genes. This should be reflected in both the foreground and background gene sets used for the enrichment tests.

As per the reviewer's comment, we have repeated pathway analysis on the DEGs after removal of all signature genes used by xCell and CIBERSORT. We found that the results do not change substantively, as the most commonly dysregulated pathways still pertain to immune signaling, immune maturation, and inflammation. This is driven in large part by the fact that the major DEGs post-filtering are still immune-related genes, some of which are immune cell-specific markers that were not present in the xCell and CIBERSORT reference panels. Nonetheless, we now more clearly identify the presence of metabolic and signaling pathways commonly dysregulated, namely the LXR/RXR and phospholipase C signaling pathways. Of note, the LXR/RXR pathway is central to fatty acid metabolism, and has been shown drive inflammation in mouse models. However, we remark that it appears these pathways are predicted to be dysregulated largely due to the presence of the post-filtered immune DEGs that act in these pathways. We have reported the full results of this new analysis in the manuscript and accompanying materials.

10. "Interestingly, recent immunological evidence suggests a clear genetic influence over tissue infiltration [7]"
Reference seems wrong

The previously cited research investigated the role of TGF- β during the earliest stages of T cell priming by generating mice lacking expression of integrin alpha-V in dendritic cells. In brief, without this gene expressed, the amount of CD8+ memory T cells in epidermis were significantly reduced and the long-term immune response could be significantly impaired. As well, previous research mentioned within that paper has shown the importance of canonical resident memory T cell markers, CD69 and alphaE-integrin, in tissue residency functions. However, to avoid confusion, we have removed this sentence and accompanying reference from the manuscript.

11. On the first results section and Figure 1: please explicitly describe the relationship between the 22 cell type panel (CSORT) and the 64 panel (xCell).

We have modified the first section of the results to more explicitly describe the commonalities and differences in cell types inferred by xCell and Cibersort:

“

To describe immune content from bulk RNA-seq samples, we used two central algorithms: xCell¹⁵ and CIBERSORT¹⁶. Both algorithms include slightly different cell types and reference gene sets for estimation. xCell relies on a modification of single sample gene-set enrichment analysis to estimate cell type scores of 64 immune and stroma cell types, including various subtypes of CD8+ T cells, CD4+ T cells, B cells, dendritic cells, macrophage polarization states, and other innate immune cells. CIBERSORT employs a linear support vector regression model to estimate cell type “relative” proportions of 22 immune cell types. This includes many of the same broad cell types as xCell, but with fewer subtypes (see **Methods** for the list of cell types estimated by xCell and CIBERSORT that were used in this manuscript). Additionally, CIBERSORT calculates a “scaling factor” to measure the amount of total immune content in the sample, allowing the calculation of “absolute” scores (which are the product of the scaling factor and cellular proportions). We refer to the relative proportions from CIBERSORT as “CIBERSORT-Relative” and the product of the relative proportions with the scaling factor as “CIBERSORT-Absolute”. We estimate three scores for each cell type to describe the immune content from the gene expression data for each tissue in each individual: xCell, CIBERSORT-Relative, and CIBERSORT-Absolute scores.

“

Within Figure caption 1b, we included additional information:

“

(b) A graphical overview of downstream statistical analysis. Immune content was estimated by using three different algorithms. xCell estimates various subtypes of CD8+ T cells, CD4+ T cells, B cells, dendritic cells, macrophage polarization states, innate immune cells, and non-immune cells. CIBERSORT, which only measures immune cells, estimates fewer subtypes and instead distinguishes between resting and activation states of major cell types. Both algorithms utilize different reference gene sets.

“

12. “Many tissues featured a majority of samples with **trace** immune enrichment,” Typo?

This sentence in the manuscript is in reference to Figure 2b, and Supplementary Figures 4-7. Across the 3 deconvolution methods, the median estimated content of CD8 T cells, macrophages, B cells, and Neutrophils in most tissues is close to zero. As a result, we reflected that the majority of samples feature “trace” enrichment in reference to the small but nonzero immune content in most samples. However, we recognize the ambiguity of the sentence as phrased, and have revised it accordingly. The current version of the manuscript states, “Many tissues featured a majority of samples with a minimal amount of immune cells...”

13. Supplementary Table 3: please specify how much variance is captured by each PC in each of the tissues.

We have added this as an additional column in Supplementary Table 3 (see Google Drive file, SuppTab3.txt): <https://drive.google.com/drive/u/1/folders/1veU3GKZqvb1mI-aH6BGPCHGuMRmyiOLr>

14. “Furthermore, we found that the top 31 iQTLs from these 31 phenotypes were significantly enriched for being a previous GWAS” please specify the fold enrichment (i.e., observed vs. expected percent of GWAS amongst the iQTLs) and P- value.

The permutation-based p-value and the observed percent of GWAS hits amongst the iQTLs are included in the current version of the paper. We have included the expected percent of GWAS hits amongst the iQTLs within our paper, which is 5.4%. This is a 3.6-fold enrichment (19.4% observed versus 5.4% expected) in our 31 observed iQTLs ($P = 5.5 \times 10^{-3}$).

15. Y axis label in Fig 5e: I am assuming the authors are referring to the T helper cells infiltration phenotype. However, “Tfh” usually stands for T follicular helper cells (a specific subtype of CD4+ T cells).

Thank you for noticing this mix-up. We have corrected the figure to now display “CD4+ helper T cells”.

References

1. Pang, W.W. *et al.* Human bone marrow hematopoietic stem cells are increased in frequency and myeloid-biased with age. *Proceedings of the National Academy of Sciences* **108**, 20012-20017 (2011).
2. Valiathan, R., Ashman, M. & Asthana, D. Effects of ageing on the immune system: infants to elderly. *Scandinavian journal of immunology* **83**, 255-266 (2016).
3. Facchini, A. *et al.* Increased number of circulating Leu 11+(CD 16) large granular lymphocytes and decreased NK activity during human ageing. *Clinical and experimental immunology* **68**, 340 (1987).
4. Amadori, A. *et al.* Genetic control of the CD4/CD8 T-cell ratio in humans. *Nature medicine* **1**, 1279 (1995).
5. Monaco, G. *et al.* RNA-Seq signatures normalized by mRNA abundance allow absolute deconvolution of human immune cell types. *Cell reports* **26**, 1627-1640. e7 (2019).
6. Sturm, G. *et al.* Comprehensive evaluation of transcriptome-based cell-type quantification methods for immuno-oncology. *Bioinformatics* **35**, i436-i445 (2019).
7. Vallania, F. *et al.* Leveraging heterogeneity across multiple datasets increases cell-mixture deconvolution accuracy and reduces biological and technical biases. *Nature communications* **9**, 1-8 (2018).
8. Pe'er, I., Yelensky, R., Altshuler, D. & Daly, M.J. Estimation of the multiple testing burden for genomewide association studies of nearly all common variants. *Genetic Epidemiology: The Official Publication of the International Genetic Epidemiology Society* **32**, 381-385 (2008).
9. Huyghe, J.R. *et al.* Discovery of common and rare genetic risk variants for colorectal cancer. *Nature genetics* **51**, 76-87 (2019).
10. Linnér, R.K. *et al.* Genome-wide association analyses of risk tolerance and risky behaviors in over 1 million individuals identify hundreds of loci and shared genetic influences. *Nature genetics* **51**, 245-257 (2019).
11. Schumacher, F.R. *et al.* Association analyses of more than 140,000 men identify 63 new prostate cancer susceptibility loci. *Nature genetics* **50**, 928 (2018).
12. Giri, A. *et al.* Trans-ethnic association study of blood pressure determinants in over 750,000 individuals. *Nature genetics* **51**, 51-62 (2019).
13. Suzuki, K. *et al.* Identification of 28 new susceptibility loci for type 2 diabetes in the Japanese population. *Nature genetics* **51**, 379-386 (2019).
14. Luciano, M. *et al.* Association analysis in over 329,000 individuals identifies 116 independent variants influencing neuroticism. *Nature genetics* **50**, 6 (2018).
15. Aran, D., Hu, Z. & Butte, A.J. xCell: digitally portraying the tissue cellular heterogeneity landscape. *Genome biology* **18**, 220 (2017).
16. Newman, A.M. *et al.* Robust enumeration of cell subsets from tissue expression profiles. *Nature methods* **12**, 453 (2015).

REVIEWERS' COMMENTS:

Reviewer #1 (Remarks to the Author):

The authors have done a nice job addressing my critiques from the prior round of review. I have just a few remaining comments (all minor) that the authors should consider addressing.

1. In at least two instances, the authors refer to the deconvolution algorithms as "single-cell" (pages 3, 13). Deconvolution methods cannot extract single-cell information from bulk tissue transcriptomes. Please clarify/correct.
2. Discussion: "these methods do not allow differentiation between tissue-resident and tissue-infiltrating cellular subsets." This is a limitation of available reference profiles, not the deconvolution algorithms. For example, if the latest version of CIBERSORT was used instead (CIBERSORTx, PMID 31061481), the authors could have separately defined reference profiles for tissue-resident and infiltrating subsets using scRNA-seq data. Please clarify this point.
3. Discussion: "While single-cell sequencing can provide a more ... accurate perspective". Perhaps, although the authors should acknowledge the caveat that scRNA-seq requires tissue dissociation if solid tissues are analyzed. As this process can severely distort cell type proportions, it is quite possible that deconvolution methods are superior for enumerating cell fractions from solid tissue biopsies when optimized reference profiles are available.

Reviewer #2 (Remarks to the Author):

The authors revised the manuscript in an effort to address many of the concerns. There are still certain issues with some of the analysis, however I believe that the paper is capable of making a useful contribution to cancer genomics.

1. Hot vs. cold analysis: the authors have improved the description of the different methods and revised the cross- tissue comparative analysis. However, I am concerned about the way hot and cold instances are being called. The source for this concern is that in the example provided by the authors (suppl figure 21) where we see that at least in one case (blood) there was *zero* overlap between cases called "hot" by the clustering method vs. the quintile method. It is not very intuitive to me how this can happen; Partial overlap is understandable; however, zero overlap casts doubt on any downstream analysis [e.g., cross tissue comparison].
2. The description of the empirical evaluation of p- value inflation is somewhat cryptic ("we sampled the covariate-adjusted phenotypic values for the helper T cell phenotype, without replacement between individuals but identically across deconvolution methods and within individuals ") Please revise and clarify.
3. Previous comment on acknowledging the simple approach (with no deconvolution) that in some instances works similarly to the deconvolution- scheme (in the case of hot vs. cold analysis, as well as the iQTL analysis, albeit with less power). In order to be completely transparent with the readers -- please include this simple analysis (hot vs. cold and iQTL) as a supplementary section and acknowledge the fact that it can recover similar trends. This comment was not addressed by the authors, yet, I believe it is important for keeping the readers informed.

We thank the reviewers for reading our manuscript and once again providing feedback. We have made revisions to our manuscript and responded to the comments, as described below. Please see tracked changes under the main manuscript text.

REVIEWERS' COMMENTS:

Reviewer #1 (Remarks to the Author):

The authors have done a nice job addressing my critiques from the prior round of review. I have just a few remaining comments (all minor) that the authors should consider addressing.

1. In at least two instances, the authors refer to the deconvolution algorithms as “single-cell” (pages 3, 13). Deconvolution methods cannot extract single-cell information from bulk tissue transcriptomes. Please clarify/correct.

We have edited the main text accordingly.

Page 3

Previous: We first hypothesized that the relative and absolute scores from CIBERSORT encapsulated different aspects of the **single-cell** deconvolution

Current: We first hypothesized that the relative and absolute scores from CIBERSORT encapsulated different aspects of the **cell-type** deconvolution

Page 11

Previous: (since differences in the sample’s **single-cell** composition will influence population-level measurements)

Current: (since differences in the sample’s **cell-type** composition will influence population-level measurements)

2. Discussion: “these methods do not allow differentiation between tissue-resident and tissue-infiltrating cellular subsets.” This is a limitation of available reference profiles, not the deconvolution algorithms. For example, if the latest version of CIBERSORT was used instead (CIBERSORTx, PMID 31061481), the authors could have separately defined reference profiles for tissue-resident and infiltrating subsets using scRNA-seq data. Please clarify this point.

This is an excellent point. We have adopted the reviewer’s recommendations and made changes accordingly:

“Furthermore, the **available reference profiles** do not allow differentiation between tissue-resident and tissue-infiltrating cellular subsets, **which would require custom reference profiles based on single-cell RNA-seq (scRNA-seq) data⁵⁷**.”

3. Discussion: “While single-cell sequencing can provide a more ... accurate perspective”. Perhaps, although the authors should acknowledge the caveat that scRNA-seq requires tissue dissociation if solid tissues are analyzed. As this process can severely distort cell type proportions, it is quite possible that deconvolution methods are superior for enumerating cell fractions from solid tissue biopsies when optimized reference profiles are available.

This is another excellent point that we now mention in the discussion of our manuscript.

“While scRNA-seq information can provide a more intricate perspective of the infiltrating immune cells, the true cell-type proportions in the samples are distorted from tissue dissociation during the sample preparation process⁵⁷. Therefore, single-cell sequencing may potentially be inferior to deconvolution for enumerating cell-type fractions from solid tissue biopsies.”

Reviewer #2 (Remarks to the Author):

The authors revised the manuscript in an effort to address many of the concerns. There are still certain issues with some of the analysis, however I believe that the paper is capable of making a useful contribution to cancer genomics.

1. Hot vs. cold analysis: the authors have improved the description of the different methods and revised the cross- tissue comparative analysis. However, I am concerned about the way hot and cold instances are being called. The source for this concern is that in the example provided by the authors (suppl figure 21) where we see that at least in one case (blood) there was *zero* overlap between cases called “hot” by the clustering method vs. the quintile method. It is not very intuitive to me how this can happen; Partial overlap is understandable; however, zero overlap casts doubt on any downstream analysis [e.g., cross tissue comparison].

We thank the reviewer for the comment. However, we believe the reviewer has referenced the incorrect figure, as supplementary figure 21 corresponds to a GeneMania network. We believe the reviewer meant to reference the previous supplementary figure 14, which contains heatmaps of “hot” patterns across individuals with both lung and whole blood samples, as derived from the clustering, quintile, and top two quintile methods. In the previous Supplementary Figure 14, the 3 heatmaps are created separately and hierarchical clustering was performed on the individuals (columns) separately for each plot. Therefore, they do not

contain the same ordering of samples across the horizontal axis. We believe that this is the source of confusion, and have modified the figure's caption accordingly:

“... Columns (individuals) were clustered by Euclidean distance. Therefore, due to differences in clustering, one column across the three plots represents three different individuals.”

Overall, there are 9 whole blood samples that are labeled “hot” in both the k-means clustering analysis and the quintiles analysis for CD8 T cells. We show this graphically, with a new Supplementary Figure 12. Supplementary Figure 12 displays a heatmap of “hot” cases for whole blood samples across the three approaches (k-means clustering, top quintile, top two quintiles), without clustering and aligned by sample (each column represents the same sample). All “hot” samples identified by the k-means approach are “hot” samples in the top quintile approach, but not vice versa.

We mention this in the Supplementary Note, “Overlap of hot clusters across clustering approaches”.

Supplementary Figure 12: “Hot” CD8 T cell cases across the whole blood samples, labelled by the three different approaches: k-means consensus clustering, top quintile, and top two quintiles. Each column represents a different sample, while each row represents a different hot-labeling approach. Red cells indicate hot cases, while blue indicates not hot cases.

2. The description of the empirical evaluation of p-value inflation is somewhat cryptic (“we sampled the covariate-adjusted phenotypic values for the helper T cell phenotype, without replacement between individuals but identically across deconvolution methods and within individuals”) Please revise and clarify.

In the supplementary section titled “**Evaluating potential EBM *P*-value inflation**”, we have revised the above description.

“In our experiment, we used the covariate-adjusted phenotype values. Each individual’s phenotype values are assigned to a new individual, such that individual *i* who is assigned the CIBERSORT-Rel value from individual *j* will also be assigned the CIBERSORT-Abs and xCell values from individual *j*. Sampling was performed without replacement. In this way, the covariance matrix between CIBERSORT-Rel, CIBERSORT-Abs, and xCell scores is preserved and identical to the original data (which is used in the Empirical Brown’s method).”

3. Previous comment on acknowledging the simple approach (with no deconvolution) that in some instances works similarly to the deconvolution- scheme (in the case of hot vs. cold analysis, as well as the iQTL analysis, albeit with less power). In order to be completely transparent with the readers -- please include this simple analysis (hot vs. cold and iQTL) as a supplementary section and acknowledge the fact that it can recover similar trends. This comment was not addressed by the authors, yet, I believe it is important for keeping the readers informed.

As stated in the previous reviewer response, we included this within the supplementary section titled “**Using an aggregate expression analysis versus deconvolution**”.